
# Impact of regional climate change and future emission scenarios on surface O₃ and PM₂.₅ over India

Matthieu Pommier[1], Hilde Fagerli[1], Michael Gauss[1], David Simpson[1,2], Sumit Sharma[3], Vinay Sinha[4],

5   Sachin D. Ghude[5], Oskar Landgren[1], Agnes Nyiri[1], Peter Wind[1]

1 Norwegian Meteorological Institute, Oslo, Norway
2 Dept. Space Sciences, Earth & Environment, Chalmers University of Technology, Gothenburg, Sweden
3 Earth Sciences and Climate Change Division, The Energy and Resources Institute (TERI), New Delhi, India
Dept. of Natural Resources, TERI University, New Delhi, India
5 Indian Institute of Tropical Meteorology, Pune, India

Correspondence to: matthieu.pommier@met.no

**Abstract.**

Eleven of the world's 20 most polluted cities are located in India and poor air quality is already a major public health issue. However, anthropogenic emissions are predicted to increase substantially in the short-term (2030) and medium-term (2050) futures in India, especially if no more policy efforts are made. In this study, the EMEP/MSC-W chemical transport model has been used to calculate changes in surface ozone (O₃) and fine particulate matter (PM₂.₅) for India in a world of changing emissions and climate. The reference scenario (for present-day) is evaluated against surface-based measurements, mainly at

urban stations. The evaluation has also been extended to other data sets which are publicly available on the web but without quality assurance. The evaluation shows high temporal correlation for O₃ (r=0.9) and high spatial correlations for PM₂.₅ (r=0.5 and r=0.8 depending on the data set) between the model results and observations. While the overall bias in PM₂.₅ is small (lower than 6%), the model overestimates O₃ by 35%. The underestimation in NOₓ titration is probably the main reason for the O₃ overestimation in the model. However, the level of agreement can be considered satisfactory in this case of a regional

model being evaluated against mainly urban measurements, and given inevitable uncertainties in much of the input data

For the 2050s, the model predicts that climate change will have distinct effects in India in terms of O₃ pollution, with a region in the North characterized by a statistically significant increase by up to 4% (2 ppb) and one in the South by a decrease up to -3% (-1.4 ppb). This variation in O₃ is found to be partly related to changes in O₃ deposition velocity caused by changes in soil moisture and, over a few areas, partly also by changes in biogenic NMVOCs.

Our calculations suggest that PM₂.₅ will increase by up to 6.5% in the 2050s, driven by increases in dust, particulate organic matter (OM) and secondary inorganic aerosols (SIA), which are mainly affected by the change in precipitation, biogenic emissions and wind speed.



The large increase in anthropogenic emissions has a larger impact than climate change, causing $O_3$ and $PM_{2.5}$ levels to increase by 13% and 67% in average in 2050s, respectively. By the 2030s, secondary inorganic aerosol is predicted to become the second largest contributor to $PM_{2.5}$ in India, and the largest in 2050s, exceeding OM and dust.







# 1. Introduction

Air pollution is a serious health concern in the world and especially over Asia (Atkinson et al., 2012). It has been identified as the fifth most important cause of mortality in India (WHO, 2014). India is one of the countries experiencing an increase in the

number of high pollution events during this last decade. With a population of 1.3 billion inhabitants, a density of 420 inhabitants per $km^2$ (12 times the population density of the United States) and a Gross domestic product (GDP) growth of 7.6% per year in 2015 (www.worldbank.org). India is one of the fastest growing economies in the world. Thus, India has to cope with many different challenges in order to continue its economic development without a negative environmental impact. Nonetheless, air pollution is progressing up in the list of policy priorities.

Heavy air pollution results from a combination of high emissions of pollutants and unfavorable weather conditions. In order to limit air pollution or to regulate the emissions of pollutants, policy measures are starting to be implemented in India at a national level (e.g. National Environment Policy, 2006: http://iced.cag.gov.in/?page_id=1037) or at city level, as in New Delhi, which banned cars with odd and even license plate numbers (UNICEF 2016 and references therein) on alternate days. In order to meet clean-air standards for reducing the public health risk and improving air quality in urban areas, the Union

Environmental Ministry of Government of India launched a national Air Quality Index as a major aggressive initiative in 2015 for air pollution mitigation (Ghude et al., 2016).

Changes in air quality are nevertheless not only driven by regulations. Climate change may also have a non-negligible impacts on air quality, by modifying atmospheric circulation (e.g. wind speed, mixing depth and transport directions), precipitations, dry deposition, emissions and the chemical production or loss rates of pollutants (e.g. Jacob and Winner, 2009). The impact of

climate change on air quality has been extensively studied in recent years with regional models (e.g. Langner et al., 2005; 2012; Hedegaard et al., 2008; Simpson et al., 2014; Trail et al., 2014; Lacressonnière et al, 2016) but to our knowledge, no study was focused on India. Climate change is however a main worry in India, especially in the occurrence and in the intensity of extreme events as floods and cyclones (e.g. Ministry of Environment and Forests, 2010; Dash et al., 2007).

Two of the main pollutants having an impact on air quality and health effects are ozone ($O_3$) and particulate matter with an

aerodynamic diameter lower than 2.5 µm ($PM_{2.5}$) (e.g. Fann et al., 2012; Lelieveld et al., 2013). Ghude et al. (2016) showed around 570 000 and 31 000 premature deaths were due to $PM_{2.5}$ and $O_3$ exposure respectively in 2011. This caused an economic cost of 640 billion USD, which is a factor of 10 higher than total expenditure on health by public and private expenditure in India.

$O_3$ is a highly oxidative pollutant formed from precursors. $O_3$ pollution mostly occurs in summer due to warmer weather

driving photochemical reactions. $O_3$ levels depend on the balance between reactive nitrogen oxide ($NO_x$) and volatile organic compounds (VOCs). In the troposphere, the main sink of $O_3$ is the reaction with the hydroxyl radical (OH) through $HO_x$



reactions (e.g. Crutzen et al., 1999). In the atmospheric boundary layer, dry deposition (uptake by the vegetation) is usually the dominant sink (e.g. Monks et al., 2015).

$O_3$ is known to be associated with respiratory morbidity and mortality (e.g. Jerrett et al., 2009; Orru et al., 2013) but $O_3$ has

increased strongly in Asia in recent decades with industrialization and urbanization (e.g. Cooper et al., 2014). Long-term exposure to high concentration of surface $O_3$ can also damage vegetation with substantial reductions in crop yields and crop quality (e.g. Ainsworth et al., 2012, Mills et al., 2011, Morgan et al., 2006). The amount of damaged crops over India is estimated at 3.5 million tons per year (Ghude et al., 2014), which would be sufficient to feed about 94 million people living below the poverty line in India.

$PM_{2.5}$ includes sulfate ($SO_4^{2-}$), nitrate ($NO_3^-$), ammonium ($NH_4^+$), organic carbon (OM), elemental carbon (EC), dust, sea salt (SS) and other compounds. Important sources of $PM_{2.5}$ emissions in India are domestic heating in winter, wood burning (mainly used for cooking), road transport with contributions from both exhaust (mostly diesel) as well as non-exhaust emissions from brake and tyre wear, and industrial combustion. $PM_{2.5}$ also includes secondary particles formed in the atmosphere from precursor gases. Secondary inorganic aerosols consist of sulphate formed from sulphur dioxide ($SO_2$) emissions, nitrate formed

from $NO_x$ emissions, and ammonium formed from ammonia ($NH_3$) emissions. Emissions of VOCs are responsible for the formation of secondary organic aerosol (SOA). The main sink of $PM_{2.5}$ is wet-deposition, associated with rain-out and wash-out by precipitation.

Long-term exposure to elevated $PM_{2.5}$ levels leads to increased risk for a variety of diseases, such as cardiovascular disease and respiratory diseases (Lim et al., 2012). The World Health Organization (WHO) states a guideline value of 10 µg/m³ annual

mean concentration (25 µg/m³ for the daily mean) that should not be exceeded in order to ensure healthy conditions. Moreover, the Global Burden of Disease (GBD) study (Forouzanfar et al., 2015) ranked exposure to $PM_{2.5}$ as the seventh most important risk factor contributing to global mortality, responsible for 2.9 million premature deaths in 2013. Nevertheless, at the country-level, India presents one of the highest population-weighted mean concentrations in the world for 2013 (Brauer et al., 2016).

This study aims to evaluate the effect of the regional climate change and future emissions change in realistic air pollutant

emission scenarios, focusing on surface $O_3$ and $PM_{2.5}$ concentrations. For this purpose, the EMEP/MSC-W chemical transport model (Sect. 2) was used, hereafter referred to as the EMEP model. In this study we conducted a 10-year simulation of air quality in India driven by downscaled meteorological fields for three periods: 2006-2015 labelled as the reference, 2026-2035 and 2045-2055. In this study, the physical and chemical processes that are responsible for the modelled changes are investigated in detail.





Section 2 describes the model set-up. Section 3 focuses on the evaluation of the reference scenario against surface-based measurements. Section 4 highlights the impact of the climate change on the level of surface $O_3$ and $PM_{2.5}$ and section 5 investigates the joint impact of the future emission scenarios. The conclusions are provided in section 6.

**2. Model set-up**

The EMEP model is a 3-D Eulerian model described in detail in Simpson et al. (2012), but for global scale modelling some

important updates since then. Most importantly, some aspects of the model's interactions between gas-phase reactions and aerosols were made, which had significant impacts on the production of $O_3$ over polluted regions in Asia and elsewhere. These updates, resulting in EMEP model version rv4.9 as used here, have been described in Simpson et al. (2016) and references cited therein. Although the model has traditionally been aimed at European simulations, global scale modelling has been possible for many years. Early results were reported e.g. in Jonson et al. (2010, 2015) and Wild et al. (2012).

The domain of each simulation covers the latitudes 5.6°N-40.7°N and the longitudes 56.2°E-101.7°E, and the horizontal resolution of the simulations follows the resolution of the meteorological data described in Section 2.1. However, the studied region is more centered over India (e.g. Fig. 4b).

Calculations of $O_3$ deposition in the EMEP model are rather detailed compared to most chemical transport models. We make use of the stomatal conductance algorithm (now commonly referred to as $DO_3SE$) originally presented in Emberson et al.

(2000, 2001), which depends on temperature, light, humidity and soil moisture. Calculation of non-stomatal sinks, in conjunction with an ecosystem specific calculation of vertical $O_3$ profiles, is an important part of this calculation as discussed in Tuovinen et al. (2004, 2009) or Simpson et al. (2003). The methodology and robustness of the calculations of $O_3$ deposition and stomatal conductance have been explored in a number of publications (Tuovinen et al. 2004, 2007, 2009, Emberson et al., 2007, Büker et al., 2012).

In this study, dust concentrations from a global simulation for 2012 (EMEP Status Report 1/2015) have been used as boundary conditions. As in the standard EMEP model, $O_3$ boundary conditions (lateral and top) are defined by the climatological $O_3$ data from Logan (1998), thus only the effect of future climate and Indian emissions are studied. The influence of the changes in inflow of $O_3$ or $PM_{2.5}$ from outside the Asian domain is not taken into account.

An initial spin-up of one year (2005) was conducted, followed by ten 1-year simulations from 2006 to 2015. This 10-yr

averaged simulation was defined as the reference. These simulations and their corresponding spin-up used the same emissions. To conduct the evaluation on the impact of future climate, similar runs were done with spin-ups of one year (2025 and 2045), followed by ten 1-year simulations from 2026 to 2035 and from 2046 to 2055, respectively. In this way, short-term (towards





2030) and medium-term (towards 2050) future climate changes have been analyzed. These short-term and medium-term Future Climate (FC) scenarios used the same anthropogenic emissions as the reference scenario.

In addition to the climate change, the impact of the future emission scenarios was investigated by using anthropogenic emissions for the 2030s and the 2050s. These simulations, referred to as Future Climate and Emissions (FCE) scenarios, were run for the same time periods as the FC scenarios, but used emissions for their respective baseline year. In order to simplify the reading, the four future scenarios are named as FC2030, FC2050, FCE2030 and FCE2050.

### 2.1. Downscaled meteorological data

In this work, the EMEP model used meteorological data from the Norwegian Earth System Model (NorESM1-M, Bentsen et al. 2013). These data were downscaled using the Weather Research and Forecasting (WRF) model version 3.4 following the RCP 8.5 scenario (Riahi et al., 2011) for the years 2006-2060. The RCP8.5 combines assumptions about high population and relatively slow income growth with modest rates of technological change and energy intensity improvements, leading in the long term to high energy demand and GHG emissions in absence of climate change policies (Riahi et al., 2011). The method

and the evaluation are further detailed in Jackson et al. (2017).

The domain used was following the CORDEX South Asia domain specifications (http://www.cordex.org/index.php?option=com_content&view=article&id=87&Itemid=614), yielding 193 by 130 grid points after removal of a 10-grid point buffer zone in each direction, on approximately 50 km horizontal resolution and with 30 vertical levels.

The different options used were Thompson microphysics, CAM radiation scheme, Noah Land-Surface Model, Mellor-Yamada-Janjic TKE scheme and Kain-Fritsch cumulus scheme. The evaluation against ERA-Interim for the temperature and APHRODITE for the precipitation, indicates that the downscaled run has a cold bias especially over the ocean, but when comparing with seven other simulations from the CORDEX South Asia ensemble it still performs among the best over the Indian subcontinent (Jackson et al., 2017). For precipitation, the monsoon season (July-September) was simulated to be too

dry, which may be at least partially caused by the too cold Indian Ocean and thus less evaporation. The Western Ghats region receives particularly little precipitation in all seasons, which can maybe be explained by the relatively coarse resolution leading to too little orographic precipitation.

### 2.2. Emissions

Anthropogenic emissions of $SO_x$, $NO_x$, CO, PM and NMVOC over India were taken from Sharma and Kumar (2016). These

data have a resolution of 36km × 36km and are available for 2011 (used for the reference, the FC2030 and the FC2050 scenarios) and for 2030 and 2050 (used for the FCE2030 and the FCE2050 scenarios, respectively).





For $NH_3$ (not available from Sharma and Kumar, 2016), and for all areas outside India, anthropogenic emissions from the ECLIPSEv5a baseline data set (http://www.iiasa.ac.at/web/home/research/researchPrograms/air/Global_emissions.html) were used (2010 for the reference, FC2030 and FC2050 scenarios; 2030 for the FCE2030 scenario; 2050 for the FCE2050 scenario).

The ECLIPSEv5a baseline emission data set was created with the GAINS model (Greenhouse gas–Air pollution Interactions and Synergies; http://www.iiasa.ac.at/web/home/research/researchPrograms/GAINS.en.html) (Amann et al., 2011), which provides emissions of long-lived greenhouse gases and shorter-lived species in a consistent framework.

The anthropogenic emissions used for India are presented in Fig. 1. These future scenarios are characterized by sharp increases in all emissions even if the CO and the $NH_3$ emissions increase somewhat less in relative terms (close to 30% by 2030 and

60% by 2050) in comparison to the other components. Indeed, the predicted increases between 2011 and 2050 are very large, amounting to 304% ($SO_x$), 287% (NMVOC), 162% ($NO_x$ and $PM_{coarse}$) and 100% ($PM_{2.5}$).

The scenario estimating the emissions used by Sharma and Kumar (2016) only incorporates the policies which were already implemented before 2014/15. Thus future road maps of stringent standards in transport and power sectors have been taken into account, but not in the industrial sector. For example, there are no standards for $NO_x$ and $SO_2$ for many coal consuming

industries. Similarly, despite reduction in biomass based combustion, there are limited controls over the fugitive NMVOC emissions which are expected to grow immensely in future. Consequently, the increase in these gases is much more than pollutants like $PM_{2.5}$, which shows much lesser increase due to interventions taken/planned by the Government of India. Although current policies have likely led to reductions in emission intensities, this may not be enough for controlling absolute emissions in future. This explains the large increase in emissions in contrast to other scenarios described for example in the

recent report from the International Energy Agency (IEA, 2016). Indeed, IEA (2016) forecasts that existing and planned policies in India will help contain pollutant emissions growth in the New Policies Scenario. Thus $SO_2$ and $NO_x$ emissions each grow by only 10% by 2040, and by 7% for the $PM_{2.5}$ emissions. In their pessimistic scenario, i.e. in the absence of policy efforts, they estimated that $SO_2$ and $PM_{2.5}$ emissions would roughly double by 2040 and $NO_x$ emissions would grow almost 2.5 times.

While the $NO_x$ and $PM_{2.5}$ emissions used hereafter follow the same trend as in the IEA report, the $SO_x$ emissions are projected to increase more, by around 4 times from 2011 to 2050. It is noteworthy there are differences in economic growth rates assumed in the IEA report and the assessments used in Sharma and Kumar (2016). Sharma and Kumar (2016) assumed higher growth rates for India than in the IEA report. This comparison shows that the emissions used in this work reflect a pessimistic scenario. The emissions will continue to grow if no stringent standards are set up and our FCE scenarios highlight the air quality issue

in India without policy effort.

For comparison, the ECLIPSEv5a emissions are also plotted in Fig. 1 since the $NH_3$ emissions from ECLIPSEv5a were used as complement of the emissions from Sharma and Kumar (2016). The emissions used in this study show larger increase, and the amount of pollutants is also higher for all compounds compared to ECLIPSEv5a, except for $NO_x$ in 2050.

For the other emissions, biogenic emissions of isoprene and monoterpene are calculated in the model by emission factors as a

function of temperature and solar radiation (Simpson et al., 2012). The land-cover data underlying these calculations are from GLC-2000 (http://bioval.jrc.ec.europa.eu/products/glc2000/glc2000.php).

The forest fire emissions used correspond to the mean of "Fire INventory from NCAR version 1.5" FINNv1.5 emissions (Wiedinmyer et al., 2014) from 2005 to 2015.

### 3. Evaluation of the reference simulation with measurements

In this section, we evaluate the levels of the simulated surface $O_3$ and $PM_{2.5}$ for the reference scenario to ensure the validity of this scenario. The pollutant concentrations were averaged over their respective decade of simulation. It is important to do this evaluation in order to identify the biases or the errors of the reference runs, in order to give confidence in the model's ability to analyze future air quality projections. It should be noted that many factors can affect such evaluations, including accuracy of the emissions, model processes, the quality of the observations, the resolution and the quality of the downscaled

meteorological fields, but good agreements found with the reference scenario increase our confidence in predicted concentrations.

### 3.1 $O_3$

Surendran et al. (2015) presented an evaluation of surface $O_3$ mixing ratios simulated by the global atmospheric chemistry and transport model MOZART-4 against surface-based measurements. We have used an updated version of this catalogue of

surface observations. In total, 22 stations were available for this comparison with different periods of measurements as shown in Fig. S1. This data set corresponds to monthly means over their corresponding period. The observations compiled by Surendran et al. (2015) are a mixture of data from the Modelling Air Pollution and Networking (MAPAN), observational network of the Ministry of Earth Sciences (MoES) and from the Indian Institute of Tropical Meteorology (IITM) over urban, suburban and rural sites, with 11, 4 and 7 stations respectively (the individual time-series are shown in Fig. S2).

Averaging the concentrations over all these sites, the simulated $O_3$ shows a high temporal correlation (r=0.9) with the data set (Fig. 2a). This shows that EMEP captures rather well the seasonal variation of the surface $O_3$ over the different sites but it overestimates the mean value. The mean overestimation is 35% (11 ppb) but it varies from site to site, between -1.4% and ca.130%. There is no clear geographical pattern of this overestimation and for the temporal correlation (Figs. 2b & 2c) but the comparison shows the lowest bias for the rural sites (15%) and the highest biases for the urban and suburban sites (Fig. 3), as





expected due to the coarse scale of the model and the titration effect discussed below. The overestimation in $O_3$ found in this

work is in agreement with previous studies (e.g. Kumar et al., 2012; Chatani et al., 2014; Sharma et al., 2016), although of

course there are many differences in both emissions and models between these studies.

Several hypotheses could explain the overestimation in monthly averaged surface $O_3$. There is very likely a misrepresentation

of the $NO_x$-$O_3$ equilibrium. Under titration conditions (typically when fresh urban NO emissions are reacting with incoming

$O_3$ to create $NO_2$) an underestimation in $NO_2$ is associated with an overestimation in $O_3$. Sharma et al. (2016) and Chatani et

al. (2014) also show overestimation in $O_3$ by the models mainly due to coarser resolutions which are not able to account for

titration chemistry at the local scales. Titration of $O_3$ with NO can occur over Indian cities (e.g. Sinha et al., 2014, Sharma et

Khare, 2017) and is difficult to reproduce in regional models (e.g. Engardt, 2008). There were unfortunately no co-located

$NO_2$ or NO measurements available for this $O_3$ data set over India. However, a comparison was attempted with $NO_2$ and $O_3$

measurements provided by https://openaq.org for 2016 over Indian cities and shown in Fig. S3. We only used sites measuring

both compounds simultaneously and continuously during all months. Moreover, https://openaq.org archives worldwide real-

time air quality measurements without validating the data. This highlights the difficulty to evaluate the model results without

reliable co-located measurements of trace gases and meteorological parameters. For India, the source of these data is the

Central Pollution Control Board of India (CPCB, http://www.cpcb.gov.in/CAAQM/frmUserAvgReportCriteria.aspx). As the

comparison with the updated version of $O_3$ data from Surendran et al. (2015), these observations reflect the $O_3$ peak around

April-May. It also illustrates the underestimation by EMEP in $NO_2$ surface concentrations and the clear overestimation in $O_3$

over urban sites. Figure S3 may also suggest that $O_x$ ($NO_2$+$O_3$) concentrations are over-predicted. As $O_x$ is conserved under

titration reactions, this suggest an overestimate of photochemical activity in the region. Some possible reasons for this might

be problems with the anthropogenic and/or biogenic emissions, or over-active chemistry, e.g. over-predictions in photolysis

rates for Indian conditions (due to a lack of aerosol effects) or problems with heterogeneous reactions. However it is important

to remind that the observations are provided without quality assurance, so data quality may also play a role.

The dilution of the urban emissions into large grid boxes for urban scale could also partly explain the overestimated $O_3$ (e.g.

Sillman et al., 1990; Pleim and Ching, 1993), especially by considering that downscaled meteorological fields were used at a

coarse resolution (50 km) for a comparison at city level. This statement needs however to be tested because an increased grid

resolution does not necessarily lead to a better simulation of $O_3$ or $NO_2$ as explained by Pleim and Ching (1993). Sharma et al.

(2017) also concluded that improving the models resolutions leads to better performance only to an extent, and may not always

show improvement with finer resolutions.

**3.2 PM$_{2.5}$**





In contrast to the $O_3$ evaluation, three different data sets were available for the evaluation of the surface $PM_{2.5}$ concentrations.

Two data sets correspond to the means over a specific period over Indian cities and are originally in-situ observations from the CPCB of India. Among these two data sets, one corresponds to the WHO database (http://www.who.int/phe/health_topics/outdoorair/databases/cities/en - database 2015). This is a database containing annual means from 2009 to 2013. The other data set corresponds to averaged concentrations over the period from 2000 to 2010 published by Dey et al. (2012). The third data set corresponds to hourly measurements at the US embassy and consulates in

India available for 2014 (i.e. over New Delhi, Chennai, Kolkata, Mumbai, Hyderabad; available on https://in.usembassy.gov/embassy-consulates/new-delhi/air-quality-data/).

As for $O_3$, this evaluation remains challenging due to the location of each site, i.e. downtown, without information about the representativeness of the measured concentrations for a larger area. Despite the difficulty of comparing urban stations with simulations from a regional model, a fair agreement (spatial correlation of 0.5 and a bias of 4%) with the data from WHO was

found with the simulated surface $PM_{2.5}$ concentrations (Fig 4a). A better agreement is found for the coastal sites, especially in the South and the East of India (Fig. 4b).

The agreement between the simulated concentrations with observations is largely improved in the comparison with the data provided in Dey et al. (2012) (Fig. 5). The correlation is around 0.8 and the bias is about -6%. It is worth noting that a few discrepancies are observed between the data sets provided by WHO and by Dey et al. (2012). For example Dey al. (2012)

presented higher concentrations for a city as Patna than the value published by WHO. Similar patterns are however noted in the measurements since a city such as Delhi is characterized by higher observed concentrations in both data sets than the value simulated by the model. This underestimation by the model can be expected given its resolution.

Despite the differences in both data sets, the comparison with the observations shows limited biases from EMEP (even though the mean normalized gross errors are large) and good correlations.

Compared to the five urban sites provided by the US Embassy and consulates, a limited agreement is found (Fig. 6) with an underestimation in $PM_{2.5}$ by EMEP for all sites, especially in winter. This comparison shows however a fair agreement especially by noting the large variability in the observations, as over New Delhi on 16 July 2014 with a $PM_{2.5}$ surface concentration ranging from 5 to 955 µg/m³. Our reference simulation has also been compared with the data provided by https://openaq.org for 2016 (Fig S.4). The observations show a large variability within each month, making the interpretation

of this comparison difficult. It is worth noting that the EMEP model predicts that a large contribution from primary particulate matter (PPM) to $PM_{2.5}$, reaching 50% in December and in January, mainly composed by primary organic matter (not shown), over the sites presented in Figs 6 and S4. There is also a main natural contribution to $PM_{2.5}$ from May to September over these sites, reaching up to 70% for dust in July for Hyderabad. A chemical speciation in the measurements will be helpful to confirm the source attribution of the $PM_{2.5}$ simulated by the EMEP model and to interpret the biases found over these cities.



Finally, it should be noted that for these simulations, the EMEP model is driven by climate-model meteorology. Such meteorology is more statistical in nature than the assimilated Numerical Weather Prediction meteorology normally used with the EMEP model, and by its nature (non-assimilated), such climate meteorology cannot reproduce actual meteorology for the periods studied. Overall, however, the results suggest that the $PM_{2.5}$ concentrations simulated by the EMEP model with this setup provide a fair representation of the surface concentrations observed at the Indian monitoring sites, even if the model

tends to underestimate the highest concentrations and overestimate the lowest ones.

**4. Impact of climate**

In this section, we analyze the differences between the FC scenarios (at short-term and medium-term, i.e. FC2030 and FC2050) and the reference scenario. All meteorological fields and pollutant concentrations were averaged over their respective decade of simulation.

**4.1 $O_3$**

The reference scenario shows large surface $O_3$ over Tibet, East India and over the Bay of Bengal along the Indian coast (Fig. 7). The large values seen over Tibet are mainly the result of topographical effects, since $O_3$ values generally increase with altitude (e.g. Loibl et al, 1994). High $O_3$ near coastal areas is also expected since the deposition velocity of $O_3$ is very low over sea (e.g. Ganzeveld et al., 2009), thus minimizing the near-surface sink which usually affects land areas.

Increased temperatures associated to climate change would be expected to coincide with a rise in surface $O_3$ due to the correlation between $O_3$ production and temperature in polluted areas as explained in Jacob and Winner (2009), although such relationships are often weak (Langner et al., 2005, 2012) and less clear in background areas. This correlation is not obvious in our simulated projections, presumably due to the large number of other factors which change, such as humidity levels, mixing heights, other meteorological changes, and biogenic emissions which are affected by climate change. As our model does not

include any $CO_2$ inhibition effect on isoprene emissions (e.g. Guenther et al., 1991; Arneth et al., 2007), these biogenic emissions are simply a function of temperature and increase in the FC scenarios but current knowledge is insufficient to make reliable predictions on this issue (Simpson et al., 2014 and references therein).

While the regions with a change in $O_3$ by using the FC2030 scenario are relatively scattered, the use of the FC2050 scenario highlights a clear North-South difference over land (Fig. 7). This is characterized by an increase in surface $O_3$ concentrations

over the Northern part of India (label (A)) by up to 4.4% (2 ppb) and a decrease over the Southern peak of India (label (B)) reaching -3.4% (-1.4 ppb) (Fig. 7). The changes are statistically significant at the 95 % level for both FC scenarios showing a robust change due to the climate change.





The correlation between the spatial change in $O_3$ ($\Delta O_3$) and $\Delta T$ over land is limited in FC2030 and FC2050 scenarios. This shows that for both FC scenarios, other processes are occurring, impacting on the thermal influence on the photochemical

production of $O_3$.

Figure 8 shows the change in one important process, the $O_3$ deposition velocity, $Vd(O_3)$. The distribution of relative difference in $O_3$ is linked to the distribution of relative difference in $Vd(O_3)$ for both FC scenarios, especially in the FC2050 scenario. Wu et al. (2012) already showed a slight increase in $O_3$ deposition in the South of India and over the Western Ghats due to an increase in the leaf area of broadleaf forests but such processes are not included in our model. Instead the changes in Vd are

due to the factors which control stomatal conductance ($g_s$) in the EMEP model, namely temperature, humidity (vapour pressure deficits), radiation, and soil moisture (Emberson et al., 2001, Simpson et al. 2012). In northern European conditions, an increase in temperature will usually result in an increase in $g_s$, but in India, temperatures are often above the optimum values, and increases in temperatures may decrease $g_s$. The other factors will also affect the sign of changes in $g_s$, such as soil moisture, shown in Fig. S5. Figure S5 shows the large impact of changes in soil moisture on the variation in $Vd(O_3)$ for both FC scenarios.

The monthly variation in soil moisture matches the variation in $Vd(O_3)$ rather well. Conversely, a relationship between the change in humidity and the change in $O_3$ or in $Vd(O_3)$ is not found (not shown).

With regard to seasonal changes and focusing on the FC2050 scenario (Fig. 9), where the signatures in the change in $O_3$ are more significant (similar plots for the FC2030 scenario shown in Fig. S6), the impact of $Vd(O_3)$ is clearly visible. Exceptions are modelled over three regions as annotated in Fig. 9, where they are numbered as (1), (2) and (3) in the distribution of the

relative differences. For these regions, the deposition velocity is correlated with the surface $O_3$, in contrast to the anti-correlation found over the rest of the domain.

During the pre-monsoon, region (1) is characterized by a high level of NMVOCs and $NO_x$. During the winter, the regions (2) and (3) are characterized by a high level of NMVOCs and a low level of $NO_x$ (Fig. 10).

During the pre-monsoon, a decrease in $NO_x$ and NMVOC is simulated over region (1) (Fig. 10). The reduction of these two

precursors may explain the decrease in $O_3$. The two other regions, regions (2) and (3) are both characterized during winter by a decrease in $NO_x$ and an increase in NMVOCs. This result gives an indication of the presence of a VOC-sensitive regime. This contrasts with the $NO_x$-sensitive regime otherwise prevailing in India as calculated by Sharma et al. (2016) and observed by Mahajan et al. (2015). It is however interesting to note that the presence of a VOC-limited regime over region (1) during the pre-monsoon and over region (2) in winter, was already observed by satellite measurements (Mahajan et al., 2015).

The NMVOCs for the reference scenario over region (3), corresponding mainly to Myanmar, are probably from biomass burning as the forest fire peak season over this region occurs in winter (e.g. Pommier et al., 2017 or van der Werf et al., 2010). For the FC2030 scenario, an identical pattern is observed with an anti-correlation between the relative difference in $O_3$ and the relative difference in $Vd(O_3)$, also with the exception of three other regions as shown in Fig. S6. This shows the change in $O_3$



is related to the change in Vd(O₃), expected over three regions, as for the FC2050 scenario. Over these three regions, the

complementary effect of NOₓ-NMVOCs is also obvious in this scenario (Fig. S7).

**4.2 PM₂.₅**

In the reference scenario, the largest surface PM₂.₅ concentrations are located over the Indo-Gangetic Plain (Fig. 11), known

to be a highly populated area (e.g. Chowdhury and Maithani 2014; or http://www.census2011.co.in/states.php) and as a large

source of pollutants emissions (e.g. Clarisse et al., 2009; Mallik and Lal 2014; Tiwari et al., 2016).

According to these calculations, climate change has a larger impact, in terms of absolute values, on PM₂.₅ than on O₃. Climate

change is predicted to lead a fairly homogeneous rise in surface PM₂.₅ levels over India, especially for the FC2050 scenario,

by up to 6.5% (4.6 µg/m³) (Fig. 11). In both FC scenarios, an increase in surface PM₂.₅ concentrations is predicted for the

Eastern part of the domain (Arabian Sea) and a decrease over the Western part of the domain (Bay of Bengal). It is worth

noting that with a mesoscale model, Glotfelty et al. (2016) also simulated an increase in PM₂.₅ over India. However, a proper

comparison with other studies remains difficult, as different models or scenarios were used. It is also noteworthy that the

changes in PM₂.₅ are statistically significant at the 95 % confidence level.

The distribution of the relative difference in PM₂.₅ is roughly homogeneous in the FC2050 scenario over India (Fig. 11) but it

does not match the pattern of precipitation change (Fig. S8). As PM₂.₅ is highly sensitive to wet scavenging, we would expect

an impact of changes in precipitation on the change in PM₂.₅, but this relationship is not shown in these distributions (Figs. 11

& S8).

The composition of PM₂.₅ is mainly dominated by dust, organic matter (OM) and secondary inorganic aerosol (SIA). SIA

includes $SO_4^{2-}$, $NO_3^-$, and $NH_4^+$. The seasonal distribution of their contribution on PM₂.₅ provides complementary information

on the composition of PM₂.₅ (Fig. S9). Generally speaking, dust dominates during the pre-monsoon and monsoon periods over

India, while the amounts of OM and SIA are large during the post-monsoon and in winter. It is also worth noting that PM₂.₅

over the Arabian Sea and Tibet are mostly influenced by dust for each season. Dust over the Arabian Sea originates from the

Sahara desert, while the Tibet plateau is a known regional source of dust (e.g. Xu et al., 2015; Xin et al., 2016). PM₂.₅ over the

Bay of Bengal is largely impacted by dust during the monsoon and OM during the winter.

The simulated OM is mainly composed of SOA. It is also interesting to note that the OM over Myanmar (region 3 in Fig. 9)

is strongly influenced by primary emissions from fires and spatially coincides with the O₃ production seen previously in Fig.

9. SOA is predicted to increase, by up to 19% for FC2030 and up to 33% for FC2050 over India. This rise is probably due to

an increase in biogenic VOCs as suggested by Heald et al., (2008) (see also Fig. 10b) as a result of temperature increases. As

noted above though, isoprene emissions might actually be inhibited by CO₂ effects in a future climate, and neither Heald's

model nor ours accounts for such effects.





In order to better interpret the seasonal process, more detailed examples over India for the FC2050 scenarios with three regions

are shown in Fig. 12. The results with the FC2030 scenario (not shown) lead to similar conclusions. The composition of $PM_{2.5}$ over these regions coincides with the overall description provided by Fig. S9, i.e. there is a large amount of dust during the pre-monsoon and the monsoon; and OM and SIA during the post-monsoon and the winter. Wind speed is also higher during the pre-monsoon and the monsoon for these three regions, explaining the large amount of dust during these seasons. The budget of dust is sensitive to precipitation while OM and SIA are also highly related to chemistry as described hereafter.

Indeed, region (1), representing mainly a rural area, is subject to a large decrease in $PM_{2.5}$ by 8% during the monsoon. This is mainly due to the reduction in dust, representing 55% of $PM_{2.5}$, largely scavenged by the increased precipitation (+36%). The increase in $PM_{2.5}$ during the pre-monsoon and during the winter is linked to the increase in dust by 15% and in OM by 10%, respectively. This increase in dust depends on the change in precipitation (10% decrease) and probably also on the increase in wind speed by 3%. The augmentation in OM is related to the increase in biogenic emissions as isoprene (+14%) and

monoterpene (+11%). During the post-monsoon, the slight rise in $PM_{2.5}$ is mainly due to the increase in OM and SIA.

The impact of dust is also still high for the region located far from the desert as region (2), but the change in the $PM_{2.5}$ level is also largely related to the change in SIA and OM in all seasons. Region (2) experiences a larger variation in $PM_{2.5}$ during the monsoon (-5%) related to the increased precipitation (+35%) and the post-monsoon (+7%) probably linked to the increase in isoprene and in monoterpene emissions (+13% and +11%, respectively). The reduction in precipitation by 25% during the pre-

monsoon probably explains the increase in $PM_{2.5}$.

For region (3), located within the Indo-Gangetic Plain, the largest variation in $PM_{2.5}$ by 20% is modeled during the post-monsoon. This shows that this region is affected by a large penalty from the climate change on surface $PM_{2.5}$ concentrations during the post-monsoon. This increase is caused by the rise in both SIA (+29%) and OM (+21%). The augmentation in SIA and OM may be related to the large increase in isoprene and in monoterpene emissions (+19% both), explained by increased

temperature. Among all the seasons and among the three selected regions, the larger increase in temperature (+0.6%) occurs in this case. It is also worth noting that it coincides with the larger growth in $O_3$ among these three regions (+6%). The changes during the pre-monsoon and the winter are mainly due to the variation in SIA, and the joint changes in SIA and OM, respectively. The decrease in $PM_{2.5}$ during the monsoon is linked to the reduction in dust and in SIA (by 5% for both), which are linked to the increase in precipitations (+16%) over this wet region (2.8 mm/day).

In addition to confirm the seasonal variation in the composition of $PM_{2.5}$ over India as shown in Fig. S9, these three cases show that the main parameters influencing the changes in the main components (SIA, OM and dust), are the precipitation, the biogenic emissions and the wind speed.

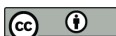


**5. Impact of future emission scenarios combined with climate change**

By combining the climate effect with future changes in emissions, we explore the differences between the FCE scenarios (2030

and 2050) and the reference scenario. As in the previous sections, the simulated fields were averaged over their respective

period of simulation.

**5.1. O$_3$**

For both FCE scenarios, a substantial increase in O$_3$ over India is modeled, as shown in Fig. 13. This augmentation in O$_3$

reaches 13% or 5 ppb in the 2030s (mean = 3% or 1 ppb) and reaching 45% or 18 ppb in the 2050s (mean = 13% or 6 ppb)

within the domain defined by the black box in Fig. 13a (latitudes 08-38°N and the longitudes 68-90°E). The increase in O$_3$ is

noticeable during the four seasons but it is more intense during the monsoon as shown by Fig. 14. It is worth noting that there

is a decrease in O$_3$ over the Western Ghats during the monsoon (e.g. region 1 in Fig. 14: -12% in 2030 and -4% in 2050) while

the rise in O$_3$ over the rest of the country is larger than for the other seasons. This contrast between the Western Ghats and the

rest of India is more pronounced in the FCE2030 scenario. Another region (numbered as 2) in winter, is also characterized by

a decrease in O$_3$ (-11% in 2030, -4% in 2050) (Fig. 14). Both reductions can be explained by the NO$_x$-VOC chemistry. Both

precursors largely increase in the FCE2030 and FCE2050 scenarios as shown by the large relative differences presented in

Fig. 15. However, regions (1) and (2) present a decrease in their NMVOC/NO$_x$ ratio in the future (Fig. 15). This ratio is already

lower in the reference scenario for both regions (≤16, Fig. 15) than in the rest of India since the mean ratio over land covering

the area defined in Fig. 13a is close to 60. This means that NO$_x$ increases more for these regions than NMVOC probably

developing a NO$_x$-saturated regime and causing the O$_3$ depletion. Thus both regions, for their respective season, suggest a

VOC-sensitive regime for the FCE2030 and FCE2050 scenarios.

**5.2. PM$_{2.5}$**

Climate change has a non-negligible impact on surface PM$_{2.5}$ concentrations, but this impact is small compared with the effects

of emissions in the FCE scenarios. Looking at the PM$_{2.5}$ in Fig. 16, a large increase is simulated throughout the domain. This

rise in surface concentrations is larger in the FCE2050 scenario than in the FCE2030 scenario. Within the region delimited by

the black box in Fig. 16a (same as Fig. 13a), the mean rise in PM$_{2.5}$ is equal to 37% (13 µg/m$^3$) in 2030s and to 67% (23 µg/m$^3$)

in 2050s. These increments alone are comparable to, or double, the annual threshold that WHO recommends not to exceed,

i.e. 10 µg/m$^3$. This increase in concentrations is also large for each season (Fig. S10). It has a maximum during the post-

monsoon in both scenarios, reaching 117% (119 µg/m$^3$) in 2030s and 172% (168µg/m$^3$) in 2050s. These huge numbers

prefigure an enormous increase in fine particulate matter over India, as already suggested by Amann et al. (2017), and imply

serious health issues for the population, especially children (UNICEF 2016).





As expected by the large increase in emissions as for SO$_x$ and NO$_x$ presented in Fig. 1, the future concentrations of PM$_{2.5}$ are influenced by $SO_4^{2-}$, $NO_3^-$, and $NH_4^+$ for each season. These compounds also show the largest increase during the post-monsoon season. This is particularly obvious for the three selected regions of Fig. 12 since SIA increases by at least 100% in

the FCE2030 scenario and by at least 200% in the FCE2050 scenario (Fig. S11). The larger increase in PM$_{2.5}$ is simulated over region (2) for both FCE scenarios during the post-monsoon; by 75% in the 2030s and 132% in the 2050s (Fig. S11). Region (3), characterized by the large impact of climate on its PM$_{2.5}$ during the post-monsoon as shown previously in Fig. 12, has an increase in PM$_{2.5}$ by around 69% in FCE2030 and 112% in FCE2050.

While the surface PM$_{2.5}$ over the land region delimited in Fig. 16a is composed on average by 5.1% of $NH_4^+$, 6.8% of $NO_3^-$,

and 9.7% of $SO_4^{2-}$ for the reference scenario; their mean contribution grows and becomes respectively 6.7%, 7.2% and 13.6% in the 2030s and 7.8%, 7.5% and 16.8% in the 2050s. OM and the dust remain two major components of surface PM$_{2.5}$ but in the 2030s, SIA becomes the second largest component since it represents 28% of PM$_{2.5}$ (29% for dust and 19% for OM) and the main component in 2050s with 32%, while dust represents 25% and OM corresponds to 18% of PM$_{2.5}$. It is also worth noting even if the PPM are high for the three scenarios (close to 20%), the amount of EC remains low, around 3%.

**6. Conclusions**

Driven by downscaled meteorological fields, the EMEP model was applied to investigate the impact of changes in regional climate and emissions on surface O$_3$ and PM$_{2.5}$ over India. The evaluation of the reference scenario with surface-based observations suggests a fair simulation of the seasonal variation of O$_3$ and a good representation of surface PM$_{2.5}$ concentrations over Indian cities. Additional information as the chemical components in PM$_{2.5}$ will be helpful to interpret the differences and

confirm the large part of primary organic matter simulated in winter by EMEP and the high ratio of dust during the monsoon. EMEP overestimates O$_3$ by 11 ppb and we suspect that NO$_x$ titration over cities, unresolved by a rather coarse grid (ca. 50 km) and possibly uncertainties in the emissions, is the main cause, especially in winter. However, there is a lack of reliable available measurements of NO$_x$ and O$_3$ to fully validate this assumption.

The O$_3$ change due to regional climate change for the medium-term (FC2050) scenario highlights a clear North-South gradient

over India, with an increase over the North, by up to 4.4% (2 ppb) and a decrease over the South, by up to -3.4% (-1.4 ppb). This O$_3$ budget is highly impacted by the change in O$_3$ deposition velocity due to the change in soil moisture, and over a few areas by the biogenic NMVOCs. Climate change leads to increases in the PM$_{2.5}$ levels at short and medium-terms, reaching 6.5% (4.6 µg/m$^3$) by the 2050s. The PM$_{2.5}$, mainly composed by dust, OM and SIA, are mainly controlled by change in precipitations and biogenic emissions. For example, over the Indo-Gangetic Plain, an increase by 20% during the post-



monsoon is predicted, related to a rise in isoprene and in monoterpene emissions, while a rural region is characterized by a 8%

decrease in $PM_{2.5}$ during the monsoon, linked to the increased precipitations in 2050.

A large increase in anthropogenic emissions is predicted if no further policy efforts are made. Combined with climate change

impacts; these emissions are predicted to lead to large changes in surface $O_3$ and $PM_{2.5}$. For surface $O_3$, these changes reach

45% over some regions in 2050. This augmentation is substantial for each season, with the exception of two regions as e.g.

over the Western Ghats during the monsoon characterized a decrease in $O_3$ around -12% in 2030 (-4% in 2050) related the

dependence of $O_3$ production on the $NO_x$ and VOC concentrations.

India is predicted to suffer large increases in $PM_{2.5}$ levels due to the increases in anthropogenic emissions in this no-further

control scenario. The increase in $PM_{2.5}$ will occur rapidly since the mean rise is close to 37% for the short-term scenario (2030s)

and 67% for the medium-term scenario (2050s). The $PM_{2.5}$ levels are predicted to reach very high levels, up to a maximum of

117% (119 µg/m$^3$) increase in 2030s and 172% (168 µg/m$^3$) in 2050s during the post-monsoon season. In 2030s, the SIA will

become the second largest component of $PM_{2.5}$ over India, exceeding the amount of OM by reaching a ratio close to 28% and

the main component in 2050s with 32%.

Finally, we can note that this is the first serious attempt to use the EMEP model over the Indian subcontinent, and there are

likely many improvements needed before modeling skill achieves the same level as obtained in European simulations. For

example, the vegetation characterization used in the EMEP model was focused on European vegetation, and is probably not

fully suitable for India, which may affect the response in temperature over India. Many issues affect any modelling study for

this region. For example, emission rates of biogenic VOC from vegetation over India are also largely unknown; Guenther et

al. (2006) show only one site in or near the Himalayas – and nothing over the rest of the Indian sub-continent. Emissions of

other compounds are also rather uncertain. Proper model evaluation in this region would require quality-assured measurements

of a range of compounds in rural as well as urban areas. Still, given the rapidly increasing emission and pollution levels in

India, it is clear that further efforts are warranted, and increasing attention will improve the basis for future model verification

and hence for a sounder basis for emissions policy assessments in future.

**Acknowledgements**

The work related to climate modelling has been supported by the Research Council of Norway through the CLIMATRANS

project (grant 235559) as well as NOTUR project `EMEP'' (nn2890k) and NorStore project ``European Monitoring and

Evaluation Programme'' (ns9005k). The EMEP project itself is supported by the Convention on the Long Range Transmission

of Air Pollutants, under UN-ECE.



**Appendix**

Error statistics used to evaluate the model performance (M and O refer, respectively, to model and observation data, and N is

the number of observations).

| Validation metrics | Formula | Range | Ideal Score |
|---|---|---|---|
| Mean Bias (MB) | $\dfrac{\sum_{i=1}^{N}(M_i - O_i)}{N}$ | -∞ to +∞ | 0 |
| Normalized Mean Bias (NMB) | $\dfrac{\sum_{i=1}^{N}(M_i - O_i)}{\sum_{i=1}^{N} O_i} \times 100\%$ | 0 to +∞ | 0 |
| Mean Normalized Gross Error (MNGE) | $\dfrac{1}{N} \sum_{i=1}^{N} \dfrac{|M_i - O_i|}{O_i} \times 100\%$ | 0 to +∞ | 0 |
| Root-Mean-Square Error (RMSE) | $\sqrt{\dfrac{\sum_{i=1}^{N}(M_i - O_i)^2}{N}}$ | 0 to +∞ | 0 |

- The MB provides the information about the absolute bias of the model, with negative values indicating
underestimation and positive values indicating overestimation by the model.

- The NMB represents the model bias relative to observations.

- The MNGE represents mean absolute difference between model and observations relative to the observations.

- The RMSE considers error compensation due to opposite sign differences and encapsulates the average error

produced by the model.

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

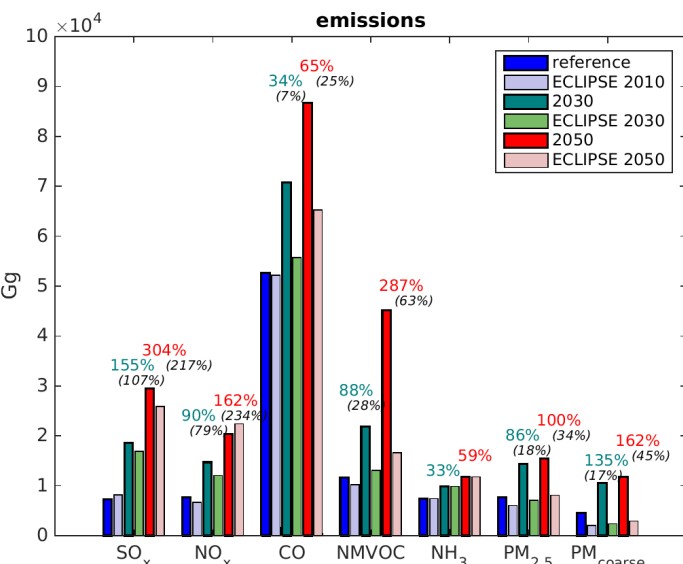

**Figure 1. Annual emissions (in Gg/yr) used for the reference (2010), FC2030 and FC2050 scenarios (dark blue), and for**
**2030 (dark green) and 2050 (dark red), used for the FCE2030 and the FCE2050 scenarios over India, respectively. The**
**variation of each compound with respect to the reference scenario is also provided by colored percent. The ECLIPSE**
**emissions are also plotted for comparison and represented by light colored bars. The variation of each compound with**
**respect to ECLIPSE2010 scenario is also provided by italic black percent given in parenthesis.**



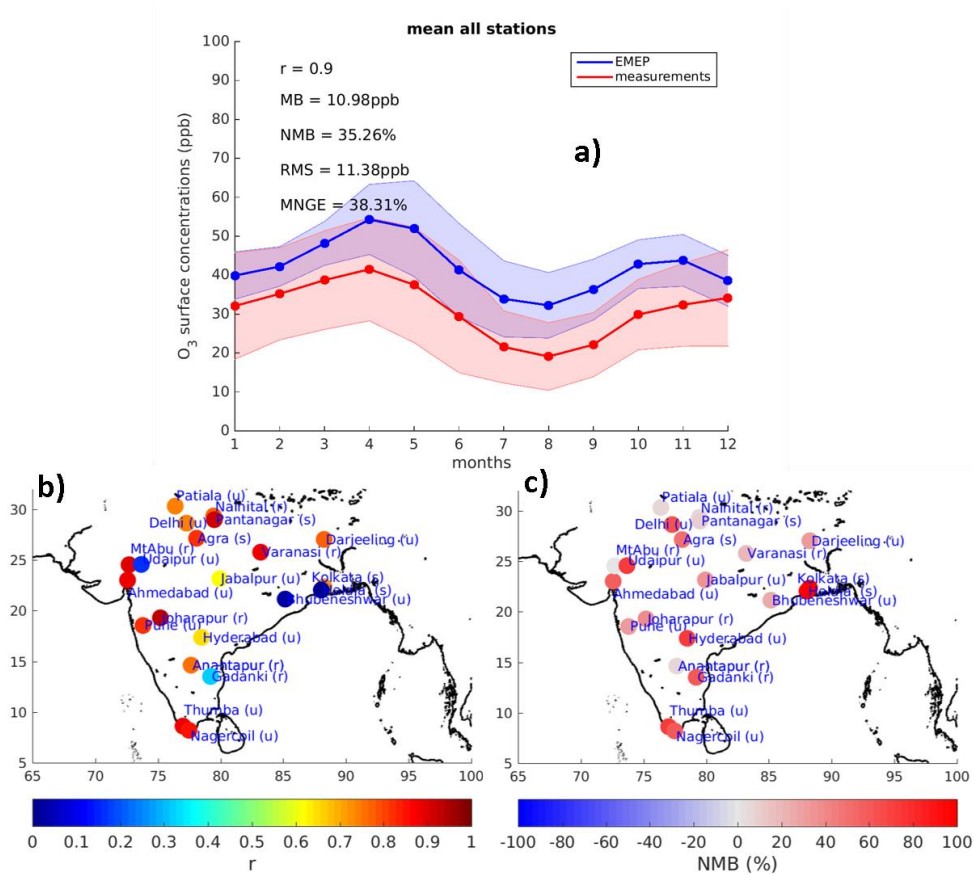

**Figure 2. a: Monthly surface O₃ mean concentrations for the 22 stations (red) and EMEP (averaged over the period of simulation) (blue). EMEP concentrations are collocated to each station. The shade error corresponds to the standard deviation. The correlation coefficient (r), the mean bias (MB), the normalized mean bias (NMB), the Root-Mean-Square (RMS) error, and the mean normalized gross error (MNGE) are provided. b: Correlation coefficient for each site. c: Normalized mean bias for each site. The type of station is given by a letter in parenthesis (u = urban, s = suburban, r = rural).**





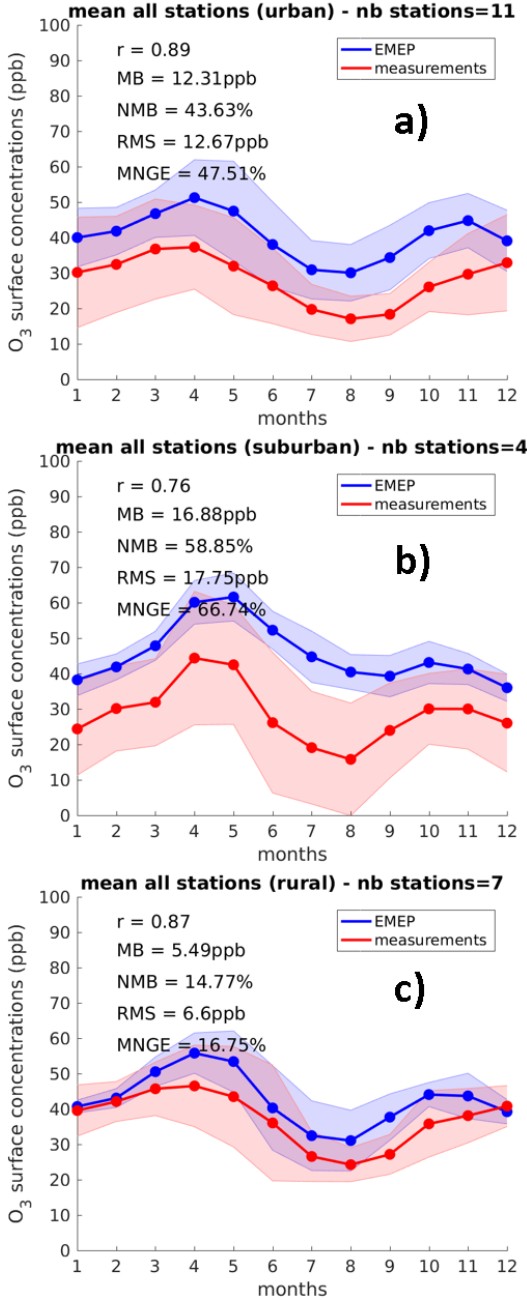

Figure 3. Monthly surface $O_3$ mean concentrations for the urban (a), suburban (b) and rural (c) stations shown in Fig. 2 (red) and EMEP (averaged over the period of simulation) (blue). EMEP concentrations are collocated to each station. The number of stations is given. The shade error corresponds to the standard deviation. The correlation coefficient (r), the mean bias (MB), the normalized mean bias (NMB), the Root-Mean-Square (RMS) error, and the mean normalized gross error (MNGE) are provided.




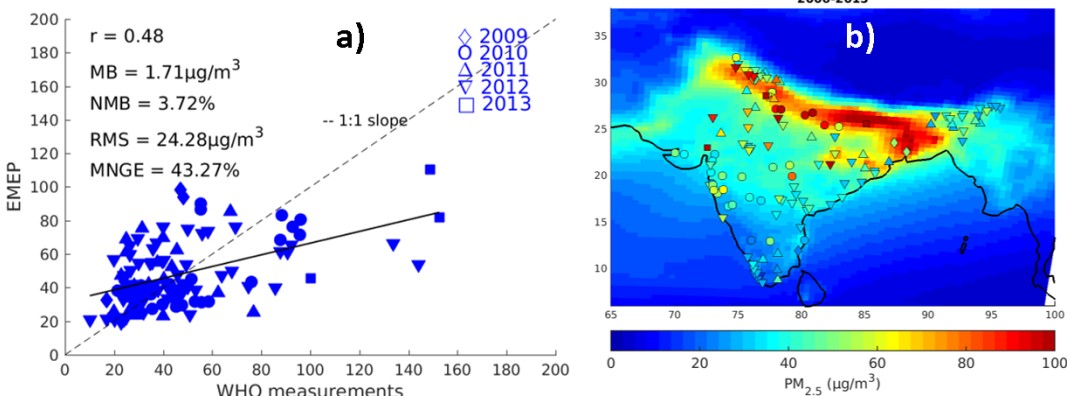

**Figure 4. a) Scatterplot between the surface PM$_{2.5}$ concentrations from EMEP (averaged over the period of simulation) and the concentrations from WHO in μg/m$^3$. Each data is represented by a different symbol for the corresponding year. The correlation coefficient (r), the mean bias (MB), the normalized mean bias (NMB), the root-mean-square (RMS) error and the mean normalized gross error (MNGE) are provided. b) Distributions of the mean surface PM$_{2.5}$ concentrations for the period 2006-2015 (reference scenario). The WHO measurements from 2009 to 2013 are superimposed on the map and represented by colored symbols following the symbols shown on the scatterplot.**


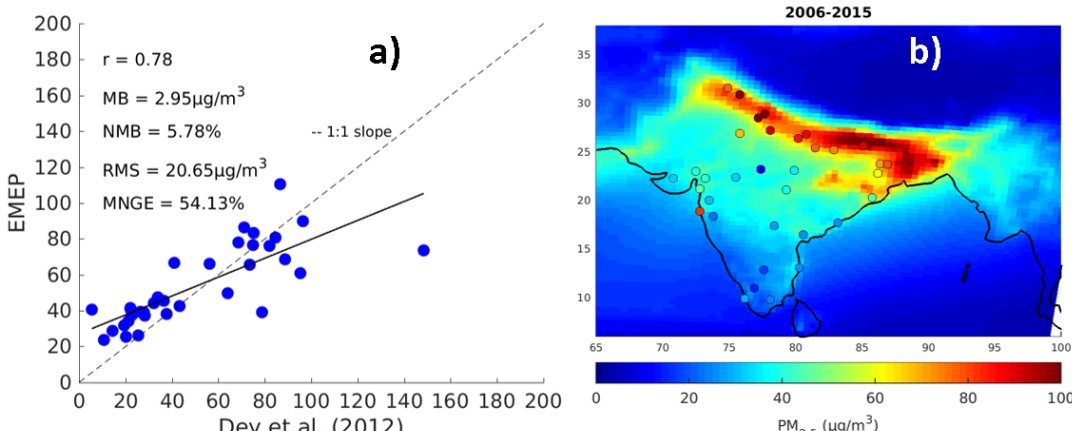

**Figure 5. a) Scatterplot between the surface PM$_{2.5}$ concentrations from EMEP (averaged over the period of simulation) and the concentrations listed in Dey et al. (2012) in μg/m$^3$. The correlation coefficient (r), the mean bias (MB), the normalized mean bias (NMB), the root-mean-square error (RMS) and the mean normalized gross error (MNGE) are provided. b) Distributions of the mean surface PM$_{2.5}$ concentrations for the period 2006-2015 (reference scenario). The measurements from Dey et al. (2012) are superimposed on the map and represented by colored dots.**






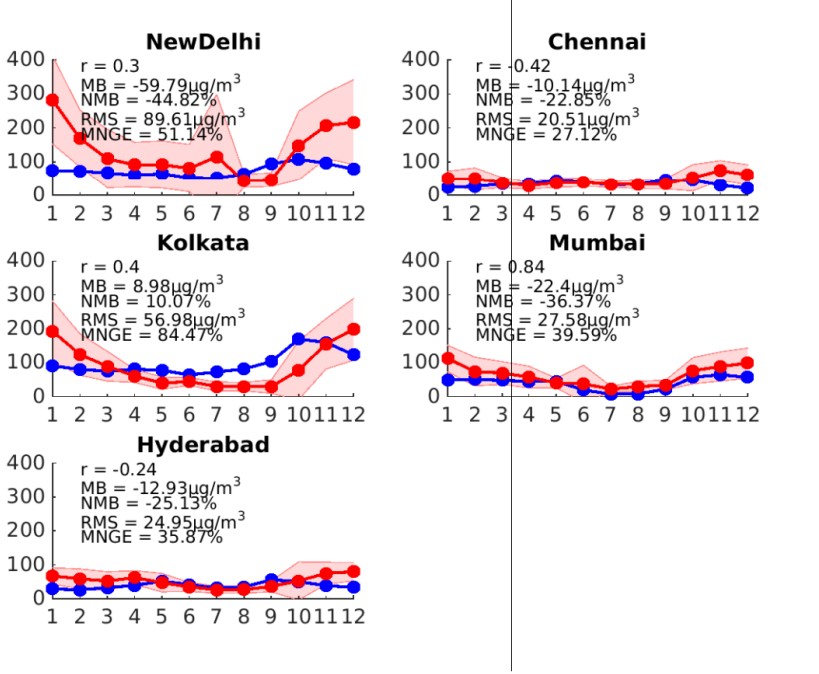

**Figure 6. Time-series of monthly surface PM$_{2.5}$ mean concentrations in µg/m³ for the observations in 2014 (red) and EMEP for the reference scenario (averaged over the period of simulation) (blue) over New Delhi, Chennai, Kolkata, Mumbai and Hyderabad. The red shade error corresponds to the standard deviation of the measurements. The correlation coefficient (r), the mean bias (MB), the normalized mean bias (NMB), the Root-Mean-Square (RMS) error, and the mean normalized gross error (MNGE) are provided.**




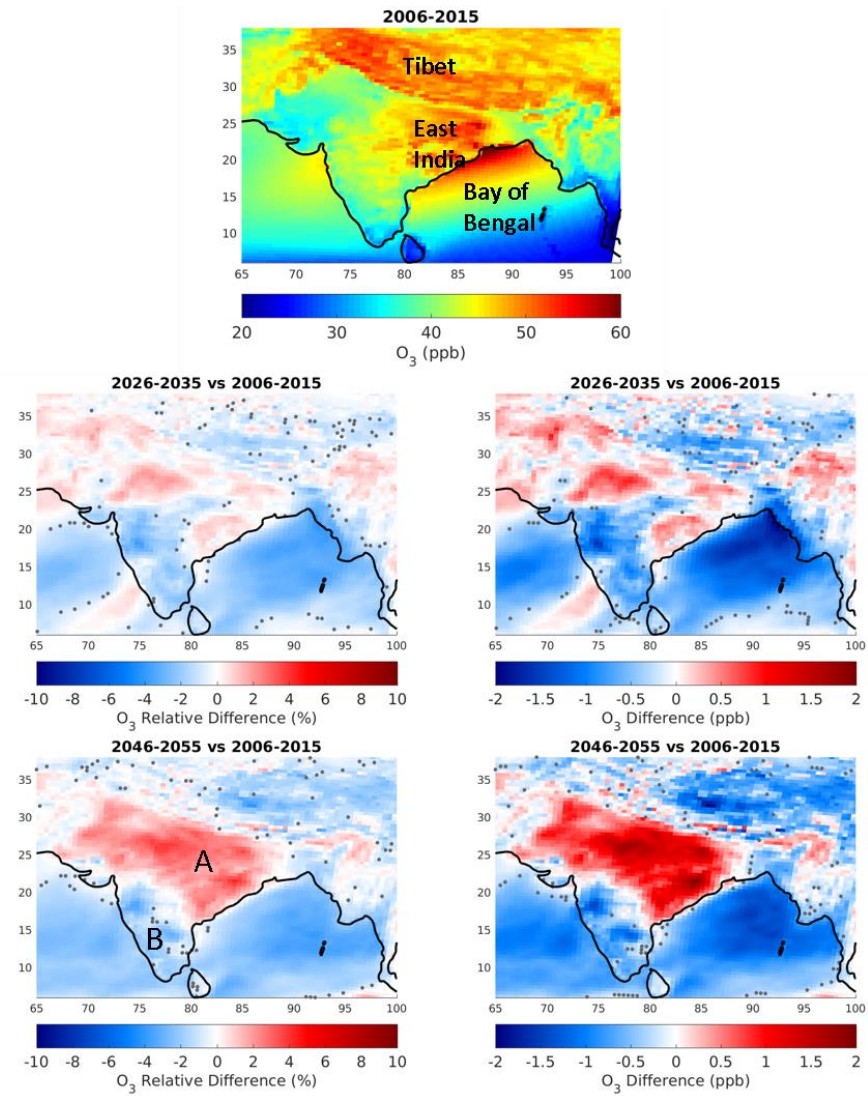

**Figure 7. Distribution of surface O₃ mixing ratios (in ppb) for the reference scenario (top panel), distribution of the relative difference and absolute difference in surface O₃ between the reference scenario and the FC2030 scenario (middle panels) and the FC2050 scenario (bottom panels). The relative differences are calculated as: ([FC – reference] / reference) × 100%, and the absolute differences as: [FC – reference]. Grey dots mark grid points that do not satisfy the 95% level of significance.**




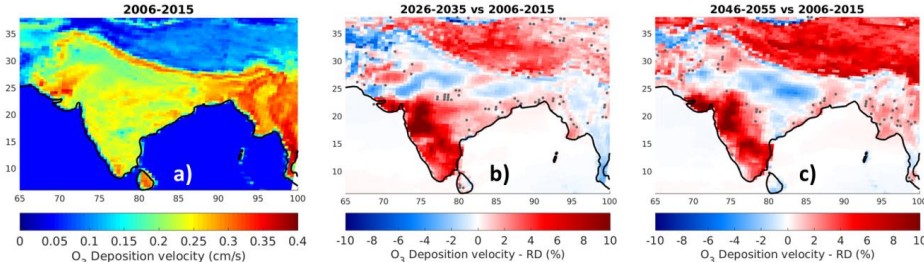

**Figure 8. Distribution of O₃ deposition velocity for the reference scenario (a) and distribution of the relative difference in O₃ deposition velocity between the reference scenario and the FC2030 scenario (b) and the FC2050 scenario (c). The relative differences are calculated as: ([FC − reference] / reference) × 100%. Grey dots mark grid points that do not satisfy the 95% level of significance.**

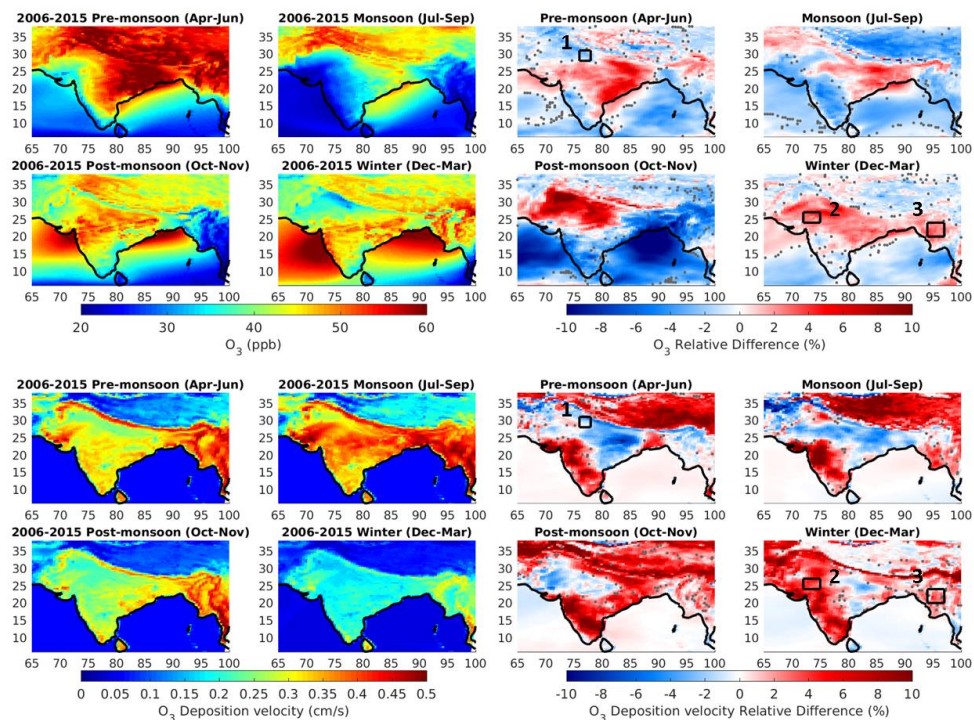

**Figure 9. Seasonal distribution of O₃ and relative difference between the reference scenario and the FC2050 scenario (top panels), seasonal distribution of O₃ deposition velocity and relative difference between the reference scenario and the FC2050 scenario (bottom panels). The relative differences are calculated as: ([FC2050 − reference] / reference) × 100%. Regions discussed in the text are numbered on the distributions of relative difference. Grey dots mark grid points that do not satisfy the 95% level of significance.**

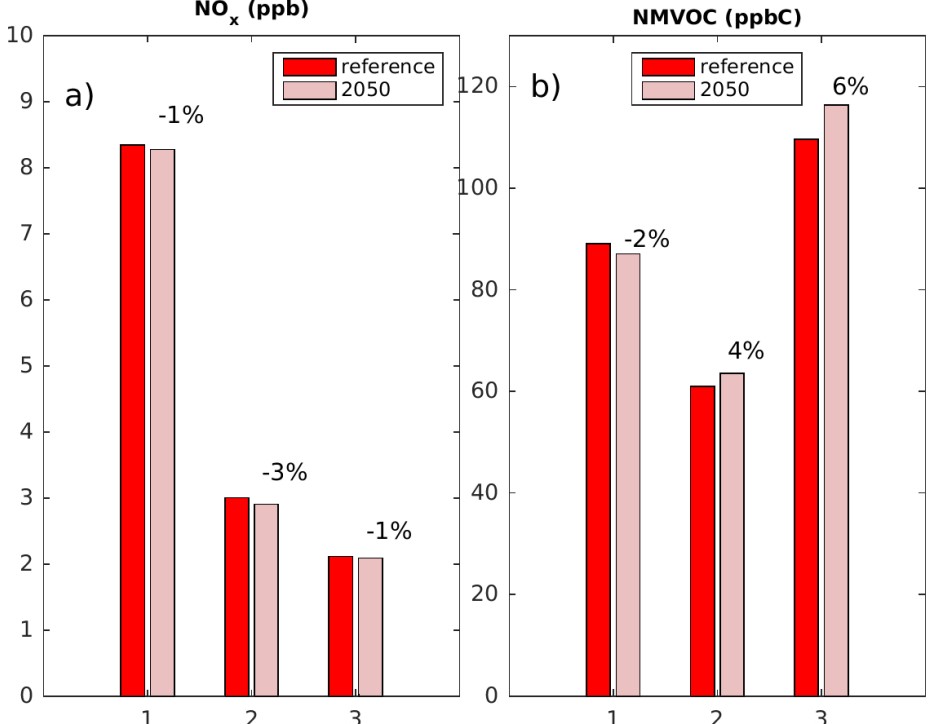

**Figure 10. Surface mixing ratios of NO$_x$ (a) and NMVOC (b) for the reference scenario (red) and the FC2050 scenario (pink) over the three selected regions during their corresponding season, highlighted in Fig. 9. The relative differences between both scenarios over each region are also indicated. The relative differences are calculated as: ([FC2050 – reference] / reference) × 100%.**




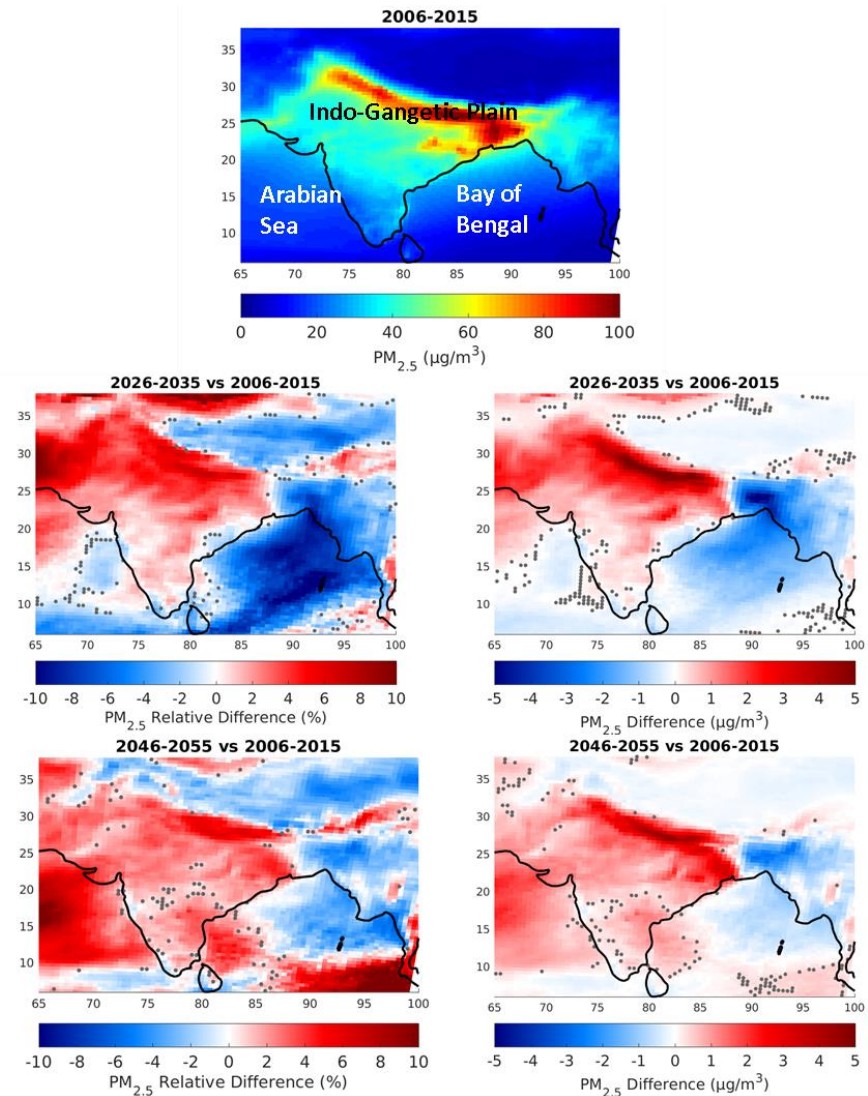

**Figure 11. Distribution of surface PM$_{2.5}$ concentrations (in μg/m$^3$) for the reference scenario (top panel), distribution of the relative difference and absolute difference in surface PM$_{2.5}$ concentrations between the reference scenario and the FC2030 scenario (middle panels) and the FC2050 scenario (bottom panels). The relative differences are calculated as: ([FC – reference] / reference) × 100%, and the absolute differences as: [FC – reference]. Grey dots mark grid points that do not satisfy the 95% level of significance.**




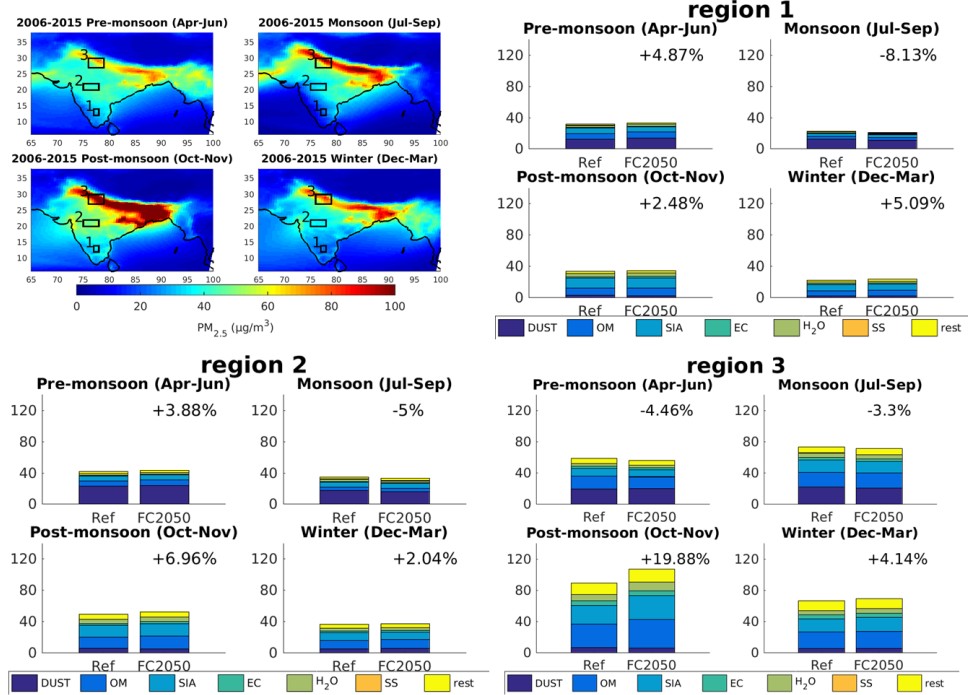

**Figure 12. Seasonal distribution of surface PM$_{2.5}$ concentrations (in µg/m$^3$) for the reference scenario, and seasonal composition of PM$_{2.5}$ (in µg/m$^3$) for the three regions highlighted by black boxes on the map for the reference and the FC2050 scenarios. The black percent corresponds to the relative difference in PM$_{2.5}$ between both scenarios for each region.**





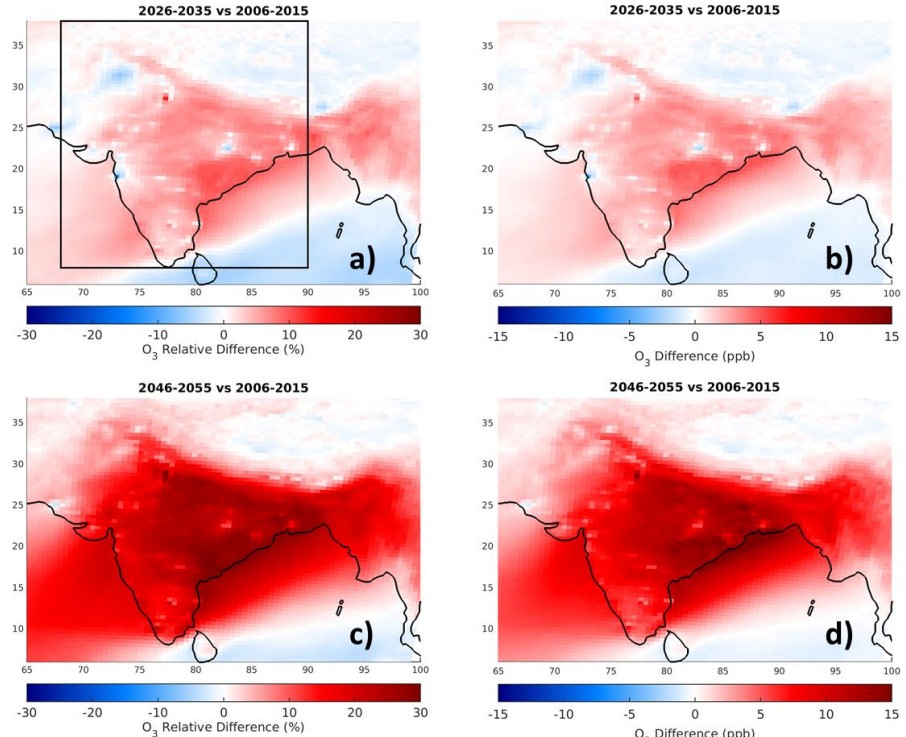

**Figure 13. Distribution of the relative difference (a and c) and absolute difference (b and d) in surface O₃ between the reference and the FCE2030 scenario (top panels) and the FCE2050 scenario (bottom panels). The relative differences are calculated as: ([FCE– reference] / reference) × 100%, and the absolute differences as: [FCE – reference]. The black box delimits the region described in the text.**






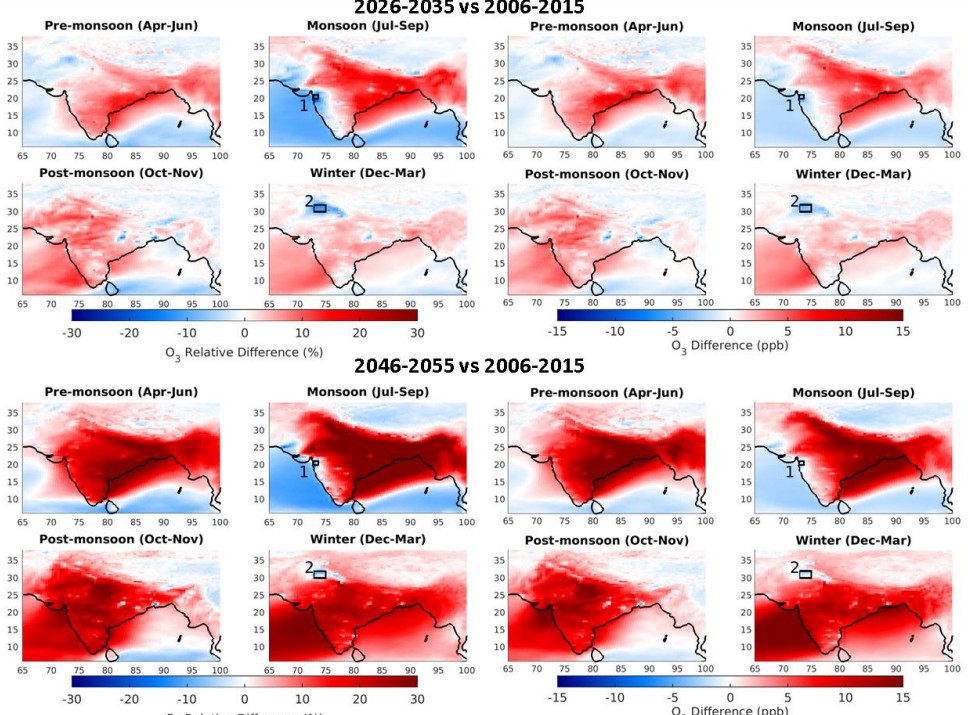

**Figure 14. Seasonal distribution of the relative difference and absolute difference in surface O₃ between the reference scenario and the FCE2030 scenario (top panels) and the FCE2050 scenario (bottom panels). The relative differences are calculated as: ([FCE – reference] / reference) × 100%, and the absolute differences as: [FCE – reference]. Regions discussed in the text are numbered on the distributions for their respective season.**






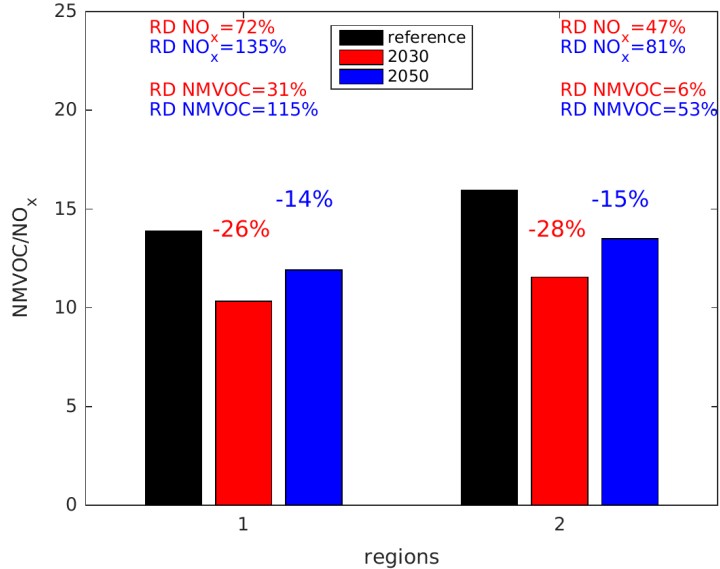

**Figure 15. Mean NMVOC/NO$_x$ ratio (ppbC/ppb) over region (1) during the monsoon and over region (2) in winter as shown in Fig. 14. The ratio for the reference scenario is plotted in black, for the FCE2030 scenario in red and for the FCE2050 scenario in blue, for both regions. The mean relative difference with respect to value in the reference scenario is written with the corresponding color. The mean relative difference for the NO$_x$ and NMVOC are also indicated.**






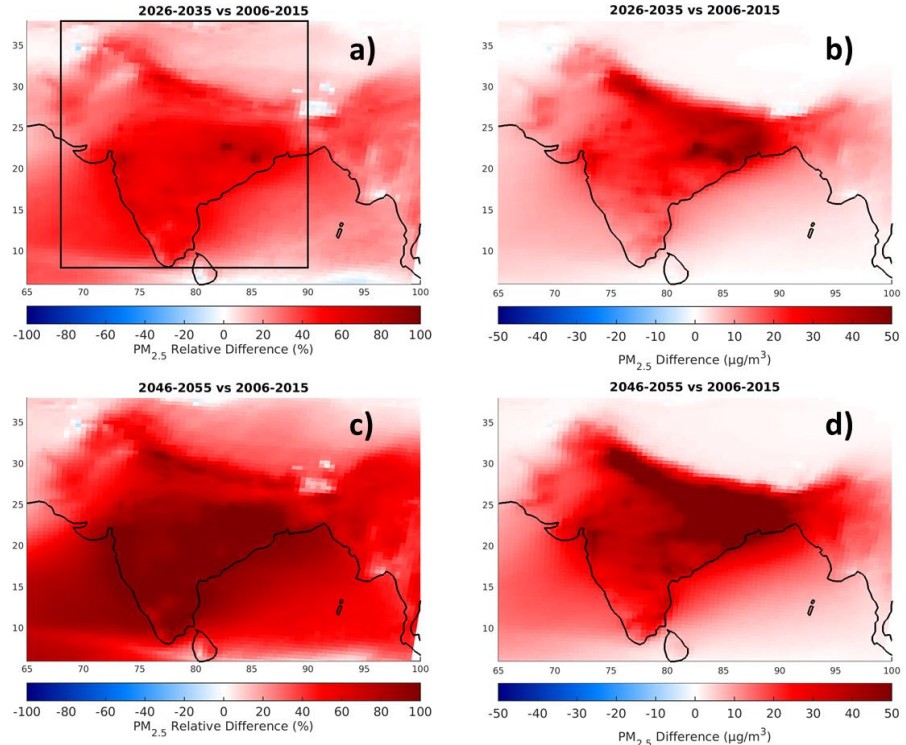

**Figure. 16. Distribution of the relative difference (a and c) and absolute difference (b and d) in surface PM$_{2.5}$ between the reference scenario and the FCE2030 scenario (top panels) and the FCE2050 scenario (bottom panels). The relative differences are calculated as: ([FCE – reference] / reference) × 100%, and the absolute differences as: [FCE – reference]. The black box delimits the region described in the text.**
