# Peer review of "Impact of regional climate change and future emission scenarios on surface $O_3$ and $PM_{2.5}$ over India"

_Atmospheric Chemistry and Physics, 2017_

## Referee Comment (RC1) · Anonymous Referee #3 · 11 Aug 2017

In this paper, the authors use the EMEP/MSC-V chemical transport model and investigate potential impact of changes in climate and emissions to 2050 on surface levels of ozone and particulate matter over the Indian sub-continent. This is the first time the EMEP model is used over this region, and the simulated present-day distribution of ozone and PM2.5 is evaluated against a range of observations. The model is then run with downscaled meteorological data and emission scenarios for 2030 and 2050.

While both climate/chemistry interactions and future pollution levels have been extensively studied, this paper contributes with additional, detailed information over a region where emissions are expected to contribute to increase strongly in the near-future. The

paper also present a useful documentation of model performance in a region where measurements have been less readily available. The paper is well-structured and well-written. I have some comments and questions for the authors to consider before the paper can be accepted, but I believe these are relatively straightforward to incorporate.

Comments:

Section 2: The model set up section should include a brief description of how aerosols are treated in the model. While possible to find in the cited literature, it will be very useful for the reader to get this information here.

Line 140: any contribution from sea salt?

Line 191: language – higher/stronger instead of much more?

Line 206: the downscaled meteorological data comes from model runs with the RCP8.5 emissions and a brief comparison of the 2030/2050 emissions in this scenario would be useful – do they differ considerably from the scenarios used in the FCE simulations?

Line 244: could the authors compare with remote sensing data? The EMEP model was part of the multi-model study by Quennehen et al (2016) – anything to learn from this? Related to this, has there been studies with EMEP over other regions that show similar problems with ozone? (e.g., Huang et al. 2017).

Line 255: "lack of aerosol effects"? Please clarify/expand.

Line 277-284: presumably there has been an increase in emission over the two different periods covered by the measurements. Could that have an impact of the comparison?

Line 295-300: even with more recent inventories, uncertainties in emissions persist, which could be worth noting/discussing.

Line 315: as well as potential changes in vegetation in a different climate.

[Figure]

Line 314-317: This is an important source of uncertainty. Are there any estimates in the literature of the potential magnitude of uncertainties introduced by this caveat? Is it large enough to affect the conclusion in this paper?

Line 323: could the authors include some information about the temperature change and its statistical significance in 2030 and 2050?

Line 353: the three regions highlighted in the 2030 results are not the same as in 2050, but are there similar explanations? Please elaborate. Could it be that the changes seen in these quite small regions are more random, than caused by the development in climate and emissions?

Line 353-354: there is something strange with this sentence. Is "expected" the right word here, or should it be "except"?

Section 4: I would like to see some discussion about the uncertainties in model representation (meteorological data) of monsoon and projected future changes, and how this could affect the results.

Line 368: see first comment – some information about how the model treats wet removal would be useful.

Line 391: changes in precipitation and wind speed will affect the dust production as well. Have the authors looked at this?

Line 401: should this be "change" instead of "variation"?

Section 5: since the motivation for this paper is partly the detrimental effects of air pollution on human health, it could be interesting to also quantify changes in terms of variables such as daily maximum 8-hour concentration if high temporal resolution model data is available and to discuss PM2.5 concentrations in terms of current air quality standards. This would make a nice, policy relevant addition. Right now, the paper focuses more on the details surrounding the smaller impact of climate change, making it somewhat unbalanced.

Line 446: please add boxes indicating regions in Fig. 16.

Line 454: Presumably, the overall PM2.5 change results in an increase in the absolute amount of EC as well, so I'm not sure about the phrasing here, i.e., "amount of EC remain low". Suggest rephrasing. Given the importance of EC/BC from a climate perspective this is an important distinction. How are PM2.5 emissions split between EC and OM in the model? (see also first comment)

Lines 449 – 454: it is interesting to note that even under increasing anthropogenic emissions, a significant fraction of PM2.5 comes from sources (dust and SOA) that are challenging, if not impossible, to control by changing policy.

References: Quennehen, B., Raut, J.-C., Law, K. S., Daskalakis, N., Ancellet, G., Clerbaux, C., Kim, S.-W., Lund, M. T., Myhre, G., Olivié, D. J. L., Safieddine, S., Skeie, R. B., Thomas, J. L., Tsyro, S., Bazureau, A., Bellouin, N., Hu, M., Kanakidou, M., Klimont, Z., Kupiainen, K., Myriokefalitakis, S., Quaas, J., Rumbold, S. T., Schulz, M., Cherian, R., Shimizu, A., Wang, J., Yoon, S.-C., and Zhu, T.: Multi-model evaluation of short-lived pollutant distributions over east Asia during summer 2008, Atmos. Chem. Phys., 16, 10765-10792, https://doi.org/10.5194/acp-16-10765-2016, 2016.

Huang, M., Carmichael, G. R., Pierce, R. B., Jo, D. S., Park, R. J., Flemming, J., Emmons, L. K., Bowman, K. W., Henze, D. K., Davila, Y., Sudo, K., Jonson, J. E., Tronstad Lund, M., Janssens-Maenhout, G., Dentener, F. J., Keating, T. J., Oetjen, H., and Payne, V. H.: Impact of intercontinental pollution transport on North American ozone air pollution: an HTAP phase 2 multi-model study, Atmos. Chem. Phys., 17, 5721-5750, https://doi.org/10.5194/acp-17-5721-2017, 2017.

---

## Referee Comment (RC2) · Anonymous Referee #1 · 11 Aug 2017

The authors used the EMEP/MSC-V model to predict the future air quality changes in India under both the climate and emission changes. The topic is not novel, especially under the RCPs scenarios. However, I do acknowledge that data analysis with high-resolution model simulations over the India are not presented frequently before.

General comments:

For the model evaluations, the authors concluded that overestimation of the ozone by 35% may be caused by the underestimation in NOX titration by the model. However, I am wondering whether the overestimation would be related to the O3 measurements the authors chose. From Fig. S1, the majority of the O3 measurement work used for the

evaluation are not adjacent to the year 2011 which was the emission year. I understand reliable observation data are scarce in India, but I presume the O3 concentration in India has been increasing for the past years. I wonder how will that affect the model evaluation performance. Please clarify.

The authors are strongly suggested to present the future climate changes in both the 2030s and 2050s, such as the temperature and precipitation. The authors discussed the effects of winds on the air pollutants. So the future changes in wind speed and directions are also necessary too.

I am not in favor of the conclusions that the O3 variations under the future climate change were caused by the O3 dry deposition changes. The authors did spend time to show the O3 dry deposition changes, but I didn't see how the authors could relate these DD changes to the O3 air quality changes. Please clarify.

I don't understand why the authors keep defining different regions for the data analysis, e.g. Figs. 7, 9, 10, 12, 13, 14. It is really not readers friendly and annoying. I have to keep going back to different figures to check which regions the authors were discussing about. I suggest the authors report air quality changes based on several larger regions consistently in the paper, or one region as the domain defined in Fig. 13.

Too many figures in the main context. I suggest move some of them to the supporting, such as Figs 8, 10, 15.

The authors should improve their writings. Lots of sentences could be combined or trimmed. I will give some examples in the specific comments.

Specific comments:

L18: change "calculate changes" to "predict changes".

L65-L66: rewrite this sentence. This is not even a complete sentence.

L77-L83: I suggest the authors to include the following two papers for summarizing the

interactions between air quality and climate change: Fiore et al., 2012, 2015.

L85: Suggest to add these two references: Silva et al., 2013; Lelieveld et al., 2015.

L94: change "but O3 has" to "and has".

L100-107: the authors should discuss more clearly about the primary PM and secondary PM as these concepts were used in the late results, otherwise it may lead to confusing. For example, in L104, the authors discussed that the "PM2.5 also includes secondary particles" which sounds to me that the authors were saying these secondary particles were at the same level as sulfate, nitrate, ammonium and so on.

L124-L128: just state the fact that this paper uses the EMEP model (rv4.9, Simpson et al., 2016), which includes some important updates such as the gas-phase reactions and aerosols compared with the previous version (Simpson et al., 2012). Discuss more in detail about the aerosol mechanisms.

L129: changed to "global scale modelling has been possible for many years (Jonson et al., 2010, 2015; Wild et al., 2015)".

L140: the author should also discuss whether the model includes the online dust module as the dust concentration would also change due to climate change too.

L144: I am confused about the model setups. So did the authors run 1-yr spin-up for each scenarios, and then run the 10 years consecutively, or did they run 1-yr spin-up for each of the 10 years simulation? "ten 1-year simulations" makes me think the authors run these 10 years simulation individually, and for each year they have their own spin-up.

L152: what does the author mean by "their respective baseline year"?

L174-L205: the authors spent great efforts to discuss the differences for the emissions between Sharma and Kumar, with ECLIPSE v5a, which makes me wonder whether the authors have chosen the best emission scenario for their simulations. Why not choose

the emission projections under the RCP8.5 which is public available and free, and also will be consistent with future climate change used in this study (Gao et al., 2012; Zhang et al., 2016).

http://tntcat.iiasa.ac.at:8787/RcpDb

The RCPs also have the NH3 emissions.

L209: delete "since the NH3 emissions from . . .."

L217: change "in order to give confidence in" to "and give confidence in"

L233: modify the "ca.130%".

L243: change "Sharma et" to "Sharma and".

L323: show the correlation for the delta O3 and delta T.

L345-L346: I am not convinced of the VOC-sensitive regime by only seeing that NOx decreases and NMVOCs increases in winter. The decreases/increases for NOx and NMVOCs are slightly (Fig. 10), and how did the authors imply there are VOC-sensitive?

L362-363: "In both FC scenarios, an increase in surface PM2.5 concentrations is predicted for the Eastern part of the domain (Arabian Sea) and a decrease over the Western part of the domain (Bay of Bengal)." I think they should be the opposite?

L436-437: "These increments alone are comparable to, or double" Rewrite this sentence with the previous one. It's really confusing.

L455: In the conclusion, the authors should discuss more about the uncertainties associated with this study, for example why the authors didn't choose the future emissions under the RCP8.5 instead of the Sharma and Kumar, 2016. How would that affect the results? This study also didn't consider the intercontinental transport of the air pollutants on the effect of surface air quality in India, which was implied to be important source in THE US (Nolte et al., 2008; Zhang et al., 2016).

L462: "emissions is the main cause" to "emissions are the main cause"

L467-L468: "Climate change leads to increases in the PM2.5 levels at short and medium-terms, reaching 6.5% (4.6 $\mu$g/m3) by the 2050s." So these "6.5%" change is regional average or domain average. It is confusing in both the abstract and conclusions since the authors keep define new regions for the analysis.

Page 37: change the colorbar for region1, region2. The fractions of the PM2.5 components were not clearly seen with the high y axis.

References:

Fiore et al., 2012, Chem Soc Rev, Global air quality and climate; doi: 10.1039/c2cs35095e.

Fiore et al., 2015, Journal of the Air & Waste Management Association, Air Quality and Climate Connections; doi: 0.1080/10962247.2015.1040526.

Gao et al., 2013, Atmos. Chem. Phys., The impact of emission and climate change on ozone in the United States under representative concentration pathways (RCPs), doi: 10.5194/acp-13-9607-2013.

Lelieveld et al., 2015, Nature, The contribution of outdoor air pollution sources to premature mortality on a global scale, doi:10.1038/nature15371.

Nolte et al., 2008, Journal of Geophysical Research, Linking global to regional models to assess future climate impacts on surface ozone levels in the United States, doi:10.1029/2007JD008497.

Silva et al., 2013, Environmental Research Letters, Global premature mortality due to anthropogenic outdoor air pollution and the contribution of past climate change, doi:10.1088/1748-9326/8/3/034005.

Zhang et al., 2016, Atmos. Chem. Phys., Co-benefits of global and regional greenhouse gas mitigation for US air quality in 2050, doi:10.5194/acp-16-9533-2016.

---

## Referee Comment (RC3) · Anonymous Referee #2 · 14 Aug 2017

The manuscript describes numerical experiments of modelling of surface O3 and PM2.5 concentrations over India using EMEP's regional off-line chemical transport model. To facilitate comparisons between present levels of air pollutants and future concentrations -after assumed changes in air pollutant emissions and in climate- the EMEP model is fed with meteorological data from a regional climate model. To my knowledge is this the first study of its kind covering the Indian subcontinent and as such the work deserves to be published.

The manuscript is well written, without any omissions and the results are, mostly, clearly presented. The manuscript could be published in its present form but it would

definitely gain from tough editing. There is an overwhelming amount of figures included in the main text which distracts the reader from any clear take-home messages. My personal feeling is that the authors want to pack too much into the present paper – which already comes with a comprehensive Supplement. The ratio between text and figures is low; chapter 5.1, for example, discusses 3 figures (altogether 21 panels) in 14 lines of text.

General comments:

Although the average seasonal cycle of O3 seems to be reasonably resolved by the EMEP model in the reference simulation (inferred by the similarity of the curves in Fig. 2a; it is not so meaningful to calculate the correlation of the 12 monthly averages of O3), is the mean bias of O3 substantial. The authors attribute this flaw to the fact that they compare the output from a regional model with observations from urban locations. I am perfectly aware of the paucity of data from regional background stations in India but the dissimilarity of station type raises concern about the validity of the model evaluation. From Fig. 3c it is clear that O3 concentrations are also overestimated during large part of the year at the available rural stations. Can the general overestimation be attributed to imperfect boundary concentrations? PM2.5 is surprisingly well reproduced by the EMEP model.

The introduction of small, rectangular, sub-regions in Fig. 9 and onwards is confusing. The selected areas don't cover all the grid-cells with the characteristics that the authors want to highlight (e.g. positive correlation between changes in O3 deposition velocity and near-surface concentration). Re-usage of the numbers 1, 2, 3 in Figs. 9, Fig. 12 and Fig. 14 further adds to the confusion. If the different sub-regions should be retained in the presentation they should be given unique numbers.

In the discussion of the results of section 5.1 and 5.2 model results have been averaged over a rectangular subdomain (shown in Figs. 13a and 16a) covering vastly different countries, socio-economical and geographical regions. I find this choice arbitrary.

To focus the presentation I would recommend the authors to consider excluding the 2026-2035 results as I don't think they add much to the general understanding of the evolution of O3 and PM2.5 from present times into the future.

Minor editorial/technical issues:

L 278: "-6%" in Fig. 5a it is +6%

L. 290-293: "It is worth nothing ... for Hyderabad." Unclear what you want to say with these sentences here.

L 363: "Eastern" and "Western" are shifted

The appendix is never mentioned in the main text. "Mean normalized Gross Error (MNGE)" is probably a valid term but I would perform the more descriptive term "Mean normalised absolute error". The formula for NMB is in error (1/N is missing).

It is unnecessary to label the increasing and decreasing O3 with A and B in Fig. 7. These areas are quite visible any way.

---

## Author Comment (AC1) · 30 Oct 2017

**Reviewer 1**

The authors used the EMEP/MSC-V model to predict the future air quality changes in India under both the climate and emission changes. The topic is not novel, especially under the RCPs scenarios. However, I do acknowledge that data analysis with high-resolution model simulations over the India are not presented frequently before.

The authors thank reviewer 1 for the thorough review. A detailed point-by-point reply (written in blue) is provided hereafter.

General comments:
For the model evaluations, the authors concluded that overestimation of the ozone by 35% may be caused by the underestimation in NOX titration by the model. However, I am wondering whether the overestimation would be related to the O3 measurements the authors chose. From Fig. S1, the majority of the O3 measurement work used for the evaluation are not adjacent to the year 2011 which was the emission year. I understand reliable observation data are scarce in India, but I presume the O3 concentration in India has been increasing for the past years. I wonder how will that affect the model evaluation performance. Please clarify.
We agree that the partial lack of coincidence between available measurements and emission data adds to the uncertainty in the model evaluation. We have added the sentence:
"The discrepancies between the periods of all the stations may have an impact on the evaluation, since the measurements do not necessarily match the emissions year used for the reference scenario."
We can note though that the overestimation is most pronounced for the urban stations (44%, Fig. 3a), as rural stations show an overestimate of only 15% (Fig. 3c). Although there are major uncertainties in the base emissions, trends, and indeed measurements, this strongly suggests that titration is a serious problem in this comparison. This may be a sub-grid problem that would require finer-scale emissions and modelling to resolve.

The authors are strongly suggested to present the future climate changes in both the 2030s and 2050s, such as the temperature and precipitation. The authors discussed the effects of winds on the air pollutants. So the future changes in wind speed and directions are also necessary too.
The changes in precipitations in the 2030s and the 2050s are already presented in fig. 8.

About the temperature, we have plotted hereafter the distribution of the temperature at 2 meter and the relative difference for both FC scenarios:

[Figure]

**Fig. 1** Distribution of the temperature at 2 meter and the relative difference for both FC scenarios. Note that the grey points, on the distribution of the relative difference, show the grids that do not satisfy the 95% level of significance.

We do not show these maps in the manuscript but we have added relevant information (highlighted in bold) in the text, in section 4.1:

"This shows that for both FC scenarios, **even though the change in temperature is statistically significant (not shown),** other processes are occurring which impact on the thermal influence on the photochemical production of $O_3$."

The maps in Fig. 2 show an increase in the wind speed over the Bay of Bengal and over a large part of the Thar Desert. There is also a decrease in the wind speed over the Indo-Gangetic Plain and the Northern part of Arabian Sea.

These changes do not match the changes in $O_3$ and $PM_{2.5}$ shown in Figs 7 and 11 of the ACPD manuscript, but we have added the following sentence (in bold) at the beginning of Section "4.2 $PM_{2.5}$":

"Climate change is predicted to lead a fairly homogeneous rise in surface $PM_{2.5}$ levels over India, especially for the FC2050 scenario, by up to 6.5% (4.6 µg/m$^3$) (Fig. 9**). This maximum increase is located over the Indo-Gangetic Plain where a decrease in surface wind speed is predicted (not shown). The decrease in wind speed may limit the emission of dust and the dispersion of the $PM_{2.5}$ emitted over this area**."

[Figure]

**Fig. 2** Distributions of surface wind speed in m/s with the wind direction for the reference scenario (left), relative difference between the FC2030 scenario and the reference scenario in % (middle), relative difference between the FC2050 scenario and the reference scenario in % (right). Note that the grey points, on the distribution of the relative difference, show the grids that do not satisfy the 95% level of significance.

The change in wind direction is small. Thus, it is not easily visible on maps. We have decided not to show the changes in the wind direction in Fig. 2, but we present hereafter the wind direction (with 10m winds) for the 3 regions presented in Fig. 12 of the ACPD version of the paper.

[Figure]

**Fig. 3** Wind rose based upon 10 m winds for the Region 1 (defined in Fig. 12 in the ACPD version) presenting the wind direction for the reference (a), FC2030 (b) and FC2050 (c) scenarios, for the 4 seasons. The colorbar shows the wind speed in m/s and the percent corresponds to the distribution of the probability of the wind speed.

[Figure]

**Fig. 4** As Fig. 3 for Region 2.

[Figure]

**Fig. 5** As Fig. 3 for Region 3.

Figs. 3-5 show the limited impact of the climate on the wind direction over the 3 regions and for each season. A small change in wind speed is observed as an increase in surface wind speed over Region 1 during the pre-monsoon and the monsoon periods, and as a decrease over Region 3 during the post-monsoon.

The increase in wind speed over region 1 was already mentioned in lines 394-395 of the manuscript. For region 3 we have added this information (in bold):

"This increase is caused by the rise in both SIA (+29%) and OM (+21%) **and probably by the reduction of the dispersion as predicted by the decrease in the surface wind speed by 5%."**

The change in surface wind speed over these 3 regions presented in Figs 3-5 is confirmed by the following seasonal maps of the relative difference:

[Figure]

**Fig. 6** Seasonal distribution of the RD (in %) in the surface wind speed between the reference scenario and the FC2050 scenario. The relative differences are calculated as: ([FC2050 − reference] / reference) × 100%.

It is important to note that the distributions in Fig. 6 do not match the change in $O_3$ presented in Fig. 9 of the ACPD manuscript but they do match the change in the $O_3$ deposition velocity presented in the same Figure.

I am not in favor of the conclusions that the O3 variations under the future climate change were caused by the O3 dry deposition changes. The authors did spend time to show the O3 dry deposition changes, but I didn't see how the authors could relate these DD changes to the O3 air quality changes. Please clarify.

We agree with the reviewer that a perfect correlation between changes in $O_3$ deposition velocity and changes in $O_3$ concentration cannot be expected, although we find good anti-correlations except in the three focus areas (labelled as 'A/B/C' in the new Fig. 8) where the changes in NMVOCs may explain the changes in $O_3$.

But it is clear that climate change will cause changes in soil moisture, and changes in soil moisture impact ozone deposition. Changes in soil moisture are thus necessarily a climate change-related factor that impacts $O_3$, although they are of course not the only one. A decrease in $O_3$ will in general not only be due to an increase in dry deposition, but it will be influenced by it. We are thus careful in stating that $O_3$ changes are only 'partly related to changes in $O_3$ deposition velocity". In the abstract we now write 'assumed to be' rather than 'found to be', based on our analysis of ozone change and dry deposition change.

Abstract: "This variation in $O_3$ is **assumed** to be partly related to changes in $O_3$ deposition velocity caused by changes in soil moisture and, over a few areas, partly also by changes in biogenic NMVOCs."

Moreover, as shown with the scatterplots in this Fig. 7, even if there are areas where the changes in $O_3$ deposition velocity and the changes in $O_3$ concentration are correlated, by choosing the model grids over land within the region 70-85E - 10-35N, there are clear anti-correlations between both parameters:

[Figure]

**Fig. 7** Scatterplot between $\Delta Vd(O_3)$ and $\Delta O_3$ over land grids for the FC2030 scenario (left panel) and the FC2050 scenario (right panel).

I don't understand why the authors keep defining different regions for the data analysis, e.g. Figs. 7, 9, 10, 12, 13, 14. It is really not readers friendly and annoying. I have to keep going back to different figures to check which regions the authors were discussing about. I suggest the authors report air quality changes based on several larger regions consistently in the paper, or one region as the domain defined in Fig. 13.

It is difficult to select the same areas for each analysis since the purpose of these distinct regions was to describe and interpret:
- the change in $O_3$ due to the climate

- the change in $PM_{2.5}$ due to the climate
- the composition of $PM_{2.5}$ and the change in $O_3$ and $PM_{2.5}$ over a larger domain for the FCE scenarios.
But we fully agree that using different regions with identical labels is confusing. We have decided to change their names to clarify our analysis in the revised manuscript.

Too many figures in the main context. I suggest move some of them to the supporting, such as Figs 8, 10, 15.
Done.

The authors should improve their writings. Lots of sentences could be combined or trimmed. I will give some examples in the specific comments.
Specific comments:
L18: change "calculate changes" to "predict changes".
It has been changed.

L65-L66: rewrite this sentence. This is not even a complete sentence.
This was a typing error. The dot located after "(www.worldbank.org)" has been replaced by a comma as below:
"With a population of 1.3 billion inhabitants, a density of 420 inhabitants per $km^2$ (12 times the population density of the United States) and a Gross domestic product (GDP) growth of 7.6% per year in 2015 (www.worldbank.org), India is one of the fastest growing economies in the world."

L77-L83: I suggest the authors to include the following two papers for summarizing the interactions between air quality and climate change: Fiore et al., 2012, 2015.
We have decided to add the reference "Fiore et al. 2015" as it seems more relevant to highlight the impact of the climate change on air quality, and not the impact of air quality on climate as also described in Fiore et al. 2012.

L85: Suggest to add these two references: Silva et al., 2013; Lelieveld et al., 2015.
These references are now added.

L94: change "but O3 has" to "and has".
Changed.

L100-107: the authors should discuss more clearly about the primary PM and secondary PM as these concepts were used in the late results, otherwise it may lead to confusing. For example, in L104, the authors discussed that the "PM2.5 also includes secondary particles" which sounds to me that the authors were saying these secondary particles were at the same level as sulfate, nitrate, ammonium and so on.
We have rephrased this part in order to clarify our description. Now it reads:
"$PM_{2.5}$ consists of both primary and secondary components. Primary $PM_{2.5}$ components include organic matter (OM), elemental carbon (EC), dust, sea salt (SS) and other compounds. Secondary $PM_{2.5}$ comprises compounds formed through atmospheric processing of gas-phase precursors. This includes various compounds as nitrate ($NO_3^-$) from $NO_x$, ammonium ($NH_4^+$) from ammonia ($NH_3$), sulphate ($SO_4^{2-}$) from sulphur dioxide ($SO_2$), and a large range of secondary organic aerosol (SOA) compounds from both anthropogenic and biogenic VOCs. Important sources of both primary and secondary $PM_{2.5}$ emissions in India are domestic heating in winter, wood burning (mainly used for cooking), road transport with contributions from both

exhaust (mostly diesel) as well as non-exhaust emissions from brake and tyre wear, and industrial combustion. The main sink of $PM_{2.5}$ is wet deposition, associated with rain-out and wash-out by precipitation."

L124-L128: just state the fact that this paper uses the EMEP model (rv4.9, Simpson et al., 2016), which includes some important updates such as the gas-phase reactions and aerosols compared with the previous version (Simpson et al., 2012). Discuss more in detail about the aerosol mechanisms.
The model description of Sect.2 has been re-written as below:

[revised manuscript text omitted]

L129: changed to "global scale modelling has been possible for many years (Jonson et al., 2010, 2015; Wild et al., 2015)".
Done

L140: the author should also discuss whether the model includes the online dust module as the dust concentration would also change due to climate change too.
Yes, there is a dust module. See our answer to your comment entitled L124-L128.

L144: I am confused about the model setups. So did the authors run 1-yr spin-up for each scenarios, and then run the 10 years consecutively, or did they run 1-yr spin-up for each of the 10 years simulation? "ten 1-year simulations" makes me think the authors run these 10 years simulation individually, and for each year they have their own spin-up.
We agree that this was confusing. Additional information has been added (in bold):

"An initial spin-up of one year (2005) was conducted, followed by ten 1-year simulations from 2006 to 2015. **Each simulation was used as spin-up of the following year of simulation. The initial spin-up (2005) was excluded from the analysis.**"

L152: what does the author mean by "their respective baseline year"?
To clarify the sentence we have added this information (in bold):
"These simulations, referred to as Future Climate and Emissions (FCE) scenarios, were run for the same time periods as the FC scenarios, but used emissions for their respective baseline year **(2030 for the 2030s and 2050 for the 2050s)**."

L174-L205: the authors spent great efforts to discuss the differences for the emissions between Sharma and Kumar, with ECLIPSE v5a, which makes me wonder whether the authors have chosen the best emission scenario for their simulations. Why not choose the emission projections under the RCP8.5 which is public available and free, and also will be consistent with future climate change used in this study (Gao et al., 2012; Zhang et al., 2016). http://tntcat.iiasa.ac.at:8787/RcpDb
The RCPs also have the NH3 emissions.
The emission estimates from Sharma and Kumar (2016) are based on local situation in India by accounting for sectoral growth rates envisaged by the Govt. of India in energy scenarios and also the interventions taken in different sectors up to 2014.
Moreover, by comparing emissions used with other studies shows closeness with other estimates, as shown in the following Table.

| Table 11.1: Comparison of Emission Inventory (Million Tonnes) in This Study with Others | | | | | | |
|---|---|---|---|---|---|---|
| Study | Year | PM10 | SO2 | NOx | NMVOC | CO |
| This study | 2011 | 10.6 | 5.6 | 7.0 | 11.4 | 46.4 |
| Garg et al. (2006) | 2005 | | 4.6 | 4.4 | | 41.7 |
| Streets et al. (2003) | 2000 | | 5.5 | 4.0 | 8.6 | 51.1 |
| Ohara et al. (2007) | 2003 | | 7.0 | 5.0 | 9.7 | 84.4 |
| Zhang et al. (2009 | 2006 | 4.0 | 5.6 | 4.9 | 10.8 | 61.1 |
| EDGAR 4.2a | 2008 | 10.9 | 8.5 | 6.4 | 10.6 | 46.3 |
| Kurokawa et al. (2013) | 2008 | 4.7 | 10.0 | 9.7 | 15.9 | 61.8 |
| Purohit et al (2010) | | 8.2 | 6.4 | 5.0 | 15.1 | |
| Lu et al. (2011) | 2008 | | 8.0 | | | |
| Klimont et al. (2009) | 2005 | | 6.4 | 5.0 | | |
| [a] http://edgar.jrc.ec.europa.eu/); | | | | | | |

Table extracted from Sharma and Kumar (2016).

In the opposite, as reviewed by Amann et al. (2013):
- The RCP scenarios include emission projections for $SO_2$, $NO_x$, VOC, BC, OC, CO, and $NH_3$, but they were not developed with a primary focus on air pollution concerns. They were developed for greenhouse gases.
The RCP scenarios employ a range of assumptions on climate policies and also assume for all countries additional control measures for air pollutants in the future beyond those currently included in national legislation.
- Thereby, these scenarios internalize additional air pollution policies, which might or might not materialize in the future. However, because the RCP scenarios explore a wide range of future climate policies, they provide indications about the impacts of GHG reductions on air pollutants.

- Scenarios that do not assume additional air pollution policies beyond current legislation indicate a potential rebound of emissions after 2030, whereas emissions decline in scenarios that assume autonomous further reductions in emission factors on the basis of the environmental Kuznets hypothesis. Thus, although air pollution might appear as a diminishing issue in the widely used RCP scenarios, this positive development will only occur if environmental policy interventions are enhanced in the future.

- Amann, M., Klimont, Z., and Wagner, F.: Regional and Global Emissions of Air Pollutants: Recent Trends and Future Scenarios, Annu. Rev. Environ. Resour., 38:31–55, doi: 10.1146/annurev-environ-052912-173303, 2013.

Moreover, in the RCP8.5 emissions inventory, only elemental carbon and organic carbon emissions are reported and not $PM_{2.5}$ and PMcoarse, as explained in your cited reference (Zhang et al., 2016).

L209: delete "since the NH3 emissions from...."
We have decided to keep this sentence since it is important to remind that the $NH_3$ emissions are from ECLIPSE. It also explains why in Fig. 1 the $NH_3$ emissions are identical.

L217: change "in order to give confidence in" to "and give confidence in"
Changed

L233: modify the "ca.130%".
It has been changed. Now it reads: "around 130%."

L243: change "Sharma et" to "Sharma and".
It has been changed.

L323: show the correlation for the delta O3 and delta T.
See for example the scatterplot for the area 70-85E, 10-35N (same region as in Fig. 7):

[Figure]

**Fig. 8** Scatterplot between $\Delta O_3$ and $\Delta T$ over land grids for the FC2030 scenario (left panel) and the FC2050 scenario (right panel).

We have noticed an error in the text. It is not "spatial" change but temporal. It has been changed in the text.

L345-L346: I am not convinced of the VOC-sensitive regime by only seeing that NOx decreases and NMVOCs increases in winter. The decreases/increases for NOx and NMVOCs are slightly (Fig. 10), and how did the authors imply there are VOC-sensitive?

We agree that our sentence was unclear. We have added the missing information (in bold) to the following sentence:

"**Combined with the increase in O₃**, this result gives an indication of the presence of a VOC-sensitive regime."

L362-363: "In both FC scenarios, an increase in surface PM2.5 concentrations is predicted for the Eastern part of the domain (Arabian Sea) and a decrease over the Western part of the domain (Bay of Bengal)." I think they should be the opposite?

Indeed. We have corrected this typing error.

L436-437: "These increments alone are comparable to, or double" Rewrite this sentence with the previous one. It's really confusing.

The sentence has been changed. Now it reads:

"These increments alone are comparable to the annual threshold that WHO recommends not to exceed, i.e. $10\,\mu g/m^3$, for the FCE2030 scenario, and the double for the FCE2050 scenario."

L455: In the conclusion, the authors should discuss more about the uncertainties associated with this study, for example why the authors didn't choose the future emissions under the RCP8.5 instead of the Sharma and Kumar, 2016. How would that affect the results? This study also didn't consider the intercontinental transport of the air pollutants on the effect of surface air quality in India, which was implied to be important source in THE US (Nolte et al., 2008; Zhang et al., 2016).

- About the choice of emissions from Sharma and Kumar (2016), see our previous response.

- The following figure shows that RCP8.5 emissions for 2010 have less NOₓ than the emissions used for the reference scenario in our work. These lower NOₓ emissions will probably lead to too much O₃ over polluted areas, and our study shows EMEP already overestimates O₃ over cities.

As previously highlighted (see our answer to the comment named L174-L205), the RCP8.5 NOₓ emissions are probably too optimistic (e.g. Amann et al., 2013) and we have preferred to rely on emissions estimates conducted by national experts.

[Figure]

**Fig. 9** Relative difference (in %) between the $NO_x$ emissions used in our work (Sharma and Kumar, (2016) over India and ECLISPE 2010 for the other countries) and the RCP8.5 (for the baseline year 2010). The relative difference is calculated as: [(our work − RCP8.5) / RCP8.5] × 100%.

- It is true that our work does not study the impact of the intercontinental transport of pollutants, as explained by the sentence "The influence of the changes in inflow of $O_3$ or $PM_{2.5}$ from outside the Asian domain is not taken into account."

We did not add the discussion in the conclusion but we have added these sentences in Section "2.2 Emissions":
"It is also interesting to note that the emissions used in the FCE scenarios are higher than the emissions used in the RCP8.5 scenarios for all species over India, except $NH_3$ (not shown). One of the drawback of these RCP8.5 emissions is that only elemental carbon and organic carbon emissions are reported and not $PM_{2.5}$ and $PM_{coarse}$ emissions (e.g. Zhang et al., 2016). Moreover, the RCP scenarios were not developed with a primary focus on air pollution concerns but for greenhouse gases (e.g. Amann et al., 2013)."

L462: "emissions is the main cause" to "emissions are the main cause"
Corrected.

L467-L468: "Climate change leads to increases in the PM2.5 levels at short and medium-terms, reaching 6.5% (4.6μg/m3) by the 2050s." So these "6.5%" change is regional average or domain average. It is confusing in both the abstract and conclusions since the authors keep define new regions for the analysis.
This number corresponds to the maximum increase in $PM_{2.5}$ over all land grids within 06-38N, 68-98E. This maximum is located over the Indo-Gangetic Plain.

We have added the information (in bold) in the abstract:
"Our calculations suggest that $PM_{2.5}$ will increase by up to 6.5% **over the Indo-Gangetic Plain in the 2050s. The increase over India is** driven by increases in dust, particulate organic matter

(OM) and secondary inorganic aerosols (SIA), which are mainly affected by the change in precipitation, biogenic emissions and wind speed.

The large increase in anthropogenic emissions has a larger impact than climate change, causing $O_3$ and $PM_{2.5}$ levels to increase by 13% and 67% in average in the 2050s **over the main part of India**, respectively."

In section 4.2:

"Climate change is predicted to lead a fairly homogeneous rise in surface $PM_{2.5}$ levels over India, especially for the FC2050 scenario, by up to 6.5% (4.6 µg/m$^3$) (Fig. 9). **This maximum increase is located over the Indo-Gangetic Plain where a decrease in surface wind speed is predicted (not shown). The decrease in wind speed may limit the emission of dust and the dispersion of the $PM_{2.5}$ emitted over this area."**

In the conclusion:

"Climate change leads to increases in the $PM_{2.5}$ levels at short and medium-terms, reaching **a maximum of** 6.5% (4.6 µg/m$^3$) **over the Indo-Gangetic Plain** by the 2050s."

Page 37: change the colorbar for region1, region2. The fractions of the PM2.5 components were not clearly seen with the high y axis.

We believe it is better to have a common colorbar for three regions to avoid confusion. However, in order to improve the reading of the fractions of the components, we have changed the y-axis for regions 1 and 2 (see below).

[Figure]

Figure 10. Seasonal distribution of surface $PM_{2.5}$ concentrations (in µg/m$^3$) for the reference scenario, and seasonal composition of $PM_{2.5}$ (in µg/m$^3$) for the three regions highlighted by black boxes on the map for the reference and the FC2050 scenarios. The black percent corresponds to the relative difference in $PM_{2.5}$ between both scenarios for each region. **Note the different y-axis for Region 3.**

---

## Author Comment (AC2) · 30 Oct 2017

**Reviewer 2**

The manuscript describes numerical experiments of modelling of surface O3 and PM2.5 concentrations over India using EMEP's regional off-line chemical transport model. To facilitate comparisons between present levels of air pollutants and future concentrations -after assumed changes in air pollutant emissions and in climate - the EMEP model is fed with meteorological data from a regional climate model. To my knowledge is this the first study of its kind covering the Indian subcontinent and as such the work deserves to be published. The manuscript is well written, without any omissions and the results are, mostly, clearly presented. The manuscript could be published in its present form but it would definitely gain from tough editing. There is an overwhelming amount of figures included in the main text which distracts the reader from any clear take-home messages. My personal feeling is that the authors want to pack too much into the present paper – which already comes with a comprehensive Supplement. The ratio between text and figures is low; chapter 5.1, for example, discusses 3 figures (altogether 21 panels) in 14 lines of text.

The authors thank the reviewer 2 for the careful reading of the manuscript and for the thorough review. A detailed point by point reply (in blue) is provided hereafter. We are aware that this manuscript contains very many figures, but we believe that most are needed in order to reinforce the points made. However, we have moved 8, 10, and 15 to the supplement.

General comments:
Although the average seasonal cycle of O3 seems to be reasonably resolved by the EMEP model in the reference simulation (inferred by the similarity of the curves in Fig. 2a; it is not so meaningful to calculate the correlation of the 12 monthly averages of O3), is the mean bias of O3 substantial. The authors attribute this flaw to the fact that they compare the output from a regional model with observations from urban locations. I am perfectly aware of the paucity of data from regional background stations in India but the dissimilarity of station type raises concern about the validity of the model evaluation.
We agree. This was the reason we decided to show the comparison site by site in Fig. S2 and we plotted the values of bias and of the correlation coefficient on maps in fig. 2b & c., showing the spatial distribution of the stations.
Even if the number of rural stations is limited, we attempted to perform such comparison with Fig. 3. Unfortunately, to our knowledge no more background stations are available.

From Fig. 3c it is clear that O3 concentrations are also overestimated during large part of the year at the available rural stations. Can the general overestimation be attributed to imperfect boundary concentrations? PM2.5 is surprisingly well reproduced by the EMEP model.
Indeed, we have added this information (in bold):
"Several hypotheses could explain the overestimation in monthly averaged surface O₃. **These include general uncertainties in anthropogenic and biogenic emissions, an overestimation in the transported O₃ from the boundary conditions (including stratospheric-tropospheric exchange), inadequate accounting for the impacts of the large PM concentrations on gas-aerosol interactions, or systematic biases in the deposition estimates**. There is **also** very likely a misrepresentation of the $NO_x$-O₃ equilibrium."

The introduction of small, rectangular, sub-regions in Fig. 9 and onwards is confusing. The selected areas don't cover all the grid-cells with the characteristics that the authors want to highlight (e.g. positive correlation between changes in O3 deposition velocity and near-surface

concentration). Re-usage of the numbers 1, 2, 3 in Figs.9, Fig.12 and Fig.14 further adds to the confusion. If the different sub-regions should be retained in the presentation they should be given unique numbers.

Our choice was to select areas for each analysis since the purpose of these distinct regions was to describe and interpret:
- the change in $O_3$ due to the climate
- the change in $PM_{2.5}$ due to the climate
- the composition of $PM_{2.5}$ and the change in $O_3$ and $PM_{2.5}$ over a larger domain for the FCE scenarios. But we agree that using different regions with identical labels was confusing. We have decided to change their names to clarify our analysis in the revised manuscript.

In the discussion of the results of section 5.1 and 5.2 model results have been averaged over a rectangular subdomain (shown in Figs. 13a and 16a) covering vastly different countries, socio-economical and geographical regions. I find this choice arbitrary.

We understand the comment from the reviewer but as you can see with the following map, the selected region covers a large part of India, which also gathers the main locations of the available observations (e.g. Figs. 2, 4-6 in the ACPD manuscript):

[Figure]

For this map, we have used another matlab file describing the borders and not only the coastlines as presented in the manuscript. In comparison with the borders shown in this map, we decided to extend our selected region up to 38N since the region between ~35-38N corresponds to Kashmir which is a region often defined as an Indian region, even if we are aware that it is an area claimed by both India and Pakistan. For the same reason, we decided not to show the borderlines. Moreover this region (between ~35-38N) is included in the emissions inventory from Sharma and Kumar (2016)

We also did not extend up to 98E in order to limit the number of grid cells over China, Bangladesh and Myanmar on the $O_3$ and $PM_{2.5}$ averages calculated within the region delimited by the black box.

We agree that other areas can be defined but we still believe that the selected region is a good representation of India. Please also note that we do not define this box as "India".

For your information, the domain used in our study is also a little bit smaller than the domain used for the air pollution forecast over India by the website IndiaAirQuality.info (see http://www.indiaairquality.info/iaqi-domain/).

To focus the presentation I would recommend the authors to consider excluding the 2026-2035 results as I don't think they add much to the general understanding of the evolution of O3 and PM2.5 from present times into the future.

The aim to present 2026-2035 was to highlight the fast impact of climate change and then the combined impact of climate and future emission scenarios on our $O_3$ and $PM_{2.5}$ distributions. We have decided to keep results for both periods.
However, the former Figs. 14 and S10 do not present the distributions for the FCE2030 scenario anymore.

Minor editorial/technical issues:

L 278: "-6%" in Fig. 5a it is +6%
Corrected.

L. 290-293: "It is worth nothing... for Hyderabad." Unclear what you want to say with these sentences here.
The sentences have been changed to clarify our explanation:
"A chemical speciation in the measurements will be helpful to interpret the biases found over these cities. Indeed, the EMEP model predicts a large contribution from primary particulate matter (PPM) to $PM_{2.5}$, reaching 50% in December and in January, mainly composed by primary organic matter (not shown), over the sites presented in Figs 6 and S4. The model also predicts a main natural contribution to $PM_{2.5}$ from May to September over these sites. For example, the site of Hyderabad reaches up to 70% in dust in July. An evaluation of the source attribution of the $PM_{2.5}$ simulated by the EMEP model will be an instructive information."

L 363: "Eastern" and "Western" are shifted
It has been corrected.

The appendix is never mentioned in the main text.
The following sentence has been added at the beginning of Section 3:
"The details of the statistical numbers are provided in the Appendix."

"Mean normalized Gross Error (MNGE)" is probably a valid term but I would perform the more descriptive term "Mean normalised absolute error".
It is correct that MNGE is a valid statistical term, which is used in numerous publications, see three examples chosen randomly:

- Kumar, R., Naja, M., Pfister, G. G., Barth, M. C., Wiedinmyer, C., and Brasseur, G. P.: Simulations over South Asia using the Weather Research and Forecasting model with Chemistry (WRF-Chem): chemistry evaluation and initial results, Geosci. Model Dev., 5, 619-648, https://doi.org/10.5194/gmd-5-619-2012, 2012
- Nguyen Thi, Kim Oanh: Integrated Air Quality Management: Asian Case Studies, March 29, 2017 by CRC Press, ISBN 9781138071841
- Qiao, X, Tang, Y,Hu, JL, Zhang, S,Li, JY,Kota, SH,Wu, L,Gao, HL,Zhang, HL,Ying, Q:Modeling dry and wet deposition of sulfate, nitrate, and ammonium ions in Jiuzhaigou National Nature Reserve, China using a source-oriented CMAQ model: Part I. Base case model results, Sc. of the total Env., 532, 831-839, DOI: 10.1016/j.scitotenv.2015.05.108, 2015.

However, we do not know and we did not find the term "Mean normalised absolute error".
We have found the normalized mean absolute error:

$$\text{NMAE} = \frac{1}{N} \frac{\sum_{i=1}^{N}|M_i - O_i|}{\max(Oi) - \min(Oi)} \times 100\%$$

See e.g.:
- Minh-Thang Do, Ted Soubdhan , Benoît Robyns: A study on the minimum duration of training data to provide a high accuracy forecast for PV generation between two different climatic zones, Renewable Energy, 85, 959-964, https://doi.org/10.1016/j.renene.2015.07.057, 2016.
- Dimitri Plemenos, Georgios Miaoulis: Intelligent Computer Graphics 2010, Springer, Berlin, Heidelberg, ISBN: 978-3-642-15689-2, DOI:https://doi.org/10.1007/978-3-642-15690-8.

Thus, we have calculated this parameter for the Figs. 2-6:
Fig2 NMAE=49.01%

Fig3a NMAE=60.85%
b NMAE=59.02%
c NMAE=25.83%

Fig.4 NMAE=12.78%

Fig5 NMAE=10.63%

Fig6
Delhi NMAE=29.78%
Chennai NMAE=31.64%
Kolkata NMAE=28.55%
Mumbai NMAE=25.83%
Hyderabad NMAE=37.71%

The formula for NMB is in error (1/N is missing).
The formula is correct:
$$NMB = \frac{(\sum_{i=1}^{N}(M_i - O_i))/N}{(\sum_{i=1}^{N}O_i)/N} \times 100\% = \frac{\sum_{i=1}^{N}(M_i - O_i)}{\sum_{i=1}^{N}O_i} \times 100\%$$
The factor 1/N is not missing.
See three examples chosen randomly providing this statistical metric:
- Qiao, X, Tang, Y,Hu, JL, Zhang, S,Li, JY,Kota, SH,Wu, L,Gao, HL,Zhang, HL,Ying, Q:Modeling dry and wet deposition of sulfate, nitrate, and ammonium ions in Jiuzhaigou National Nature Reserve, China using a source-oriented CMAQ model: Part I. Base case model results, Sc. of the total Env., 532, 831-839, DOI: 10.1016/j.scitotenv.2015.05.108, 2015.
- http://www.ecd.bnl.gov/steve/pres/metrics.pdf and
- Lucjan Pawlowski, Marzenna R. Dudzinska, Artur Pawlowski: Environmental Engineering III, March 23, 2010 by CRC Press, ISBN 9780415548823.

It is unnecessary to label the increasing and decreasing O3 with A and B in Fig. 7. These areas are quite visible any way
The labels have been deleted.

---

## Author Comment (AC3) · 30 Oct 2017

**Reviewer 3**

In this paper, the authors use the EMEP/MSC-V chemical transport model and investigate potential impact of changes in climate and emissions to 2050 on surface levels of ozone and particulate matter over the Indian sub-continent. This is the first time the EMEP model is used over this region, and the simulated present-day distribution of ozone and PM2.5 is evaluated against a range of observations. The model is then run with downscaled meteorological data and emission scenarios for 2030 and 2050.

While both climate/chemistry interactions and future pollution levels have been extensively studied, this paper contributes with additional, detailed information over a region where emissions are expected to contribute to increase strongly in the near-future. The paper also present a useful documentation of model performance in a region where measurements have been less readily available. The paper is well-structured and well-written. I have some comments and questions for the authors to consider before the paper can be accepted, but I believe these are relatively straightforward to incorporate.

The authors would like to thank the reviewer 3 for the detailed comments, which help to improve the manuscript. We have tried to clarify the points raised by the reviewer and to answer all remarks. Our responses are written in blue in this document. Please note that Figs 8, 10, and 15 have been moved to the supplement. Furthermore, the former Figs. 14 and S10 do not present the distributions for the FCE2030 scenario anymore.

Comments:
Section 2: The model set up section should include a brief description of how aerosols are treated in the model. While possible to find in the cited literature, it will be very useful for the reader to get this information here.
The model description of Sect.2 has been re-written as below:

[revised manuscript text omitted]

Line 191: language – higher/stronger instead of much more?

It has been changed to "larger".

Line 206: the downscaled meteorological data comes from model runs with the RCP8.5 emissions and a brief comparison of the 2030/2050 emissions in this scenario would be useful – do they differ considerably from the scenarios used in the FCE simulations?

The following maps, representing the relative difference between the emissions used for our FCE scenarios and the RCP8.5 emissions for the corresponding year, show the larger increase in the emissions compared to RCP. Only the $NH_3$ emissions, which are from ECLIPSE, are lower than the RCP8.5 emissions.

In the RCP8.5 emissions inventory, only elemental carbon and organic carbon emissions are reported and not $PM_{2.5}$ and PMcoarse, as explained in Zhang et al. (2016). Moreover, as described in Amann et al. (2013), the RCP scenarios were mostly designed for greenhouse gases, they were not developed with a primary focus on air pollution concerns.

The RCP scenarios employ a range of assumptions on climate policies and also assume for all countries additional control measures for air pollutants in the future beyond those currently included in national legislation.

Amann, M., Klimont, Z., and Wagner, F.: Regional and Global Emissions of Air Pollutants: Recent Trends and Future Scenarios, Annu. Rev. Environ. Resour., 38:31–55, doi: 10.1146/annurev-environ-052912-173303, 2013.

Zhang et al., 2016, Atmos. Chem. Phys., Co-benefits of global and regional greenhouse gas mitigation for US air quality in 2050, doi:10.5194/acp-16-9533-2016.

[Figure]

[Figure]

[Figure]

[Figure]

**Fig.** Relative difference (in %) between the NMVOC, NH$_3$, SO$_x$ and NO$_x$ emissions used in our work for the FCE2030 scenario and the RCP8.5 (for the baseline year 2030). The relative difference is calculated as: [(our work − RCP8.5) / RCP8.5 ] × 100%.

[Figure]

**Fig.** Relative difference (in %) between the NMVOC, NH$_3$, SO$_x$ and NO$_x$ emissions used in our work for the FCE2050 scenario and the RCP8.5 (for the baseline year 2050). The relative difference is calculated as: [(our work − RCP8.5) / RCP8.5 ] × 100%.

We have added these sentences:
"It is also interesting to note that the emissions used in the FCE scenarios are higher than the emissions used in the RCP8.5 scenarios for all species over India, except NH$_3$ (not shown). One of the drawback of these RCP8.5 emissions is that only elemental carbon and organic carbon emissions are reported and not PM$_{2.5}$ and PMc$_{oarse}$ emissions (e.g. Zhang et al., 2016). Moreover, the RCP scenarios were not developed with a primary focus on air pollution concerns but for greenhouse gases (e.g. Amann et al., 2013)."

Line 244: could the authors compare with remote sensing data? The EMEP model was part of the multi-model study by Quennehen et al (2016) – anything to learn from this? Related to this,

has there been studies with EMEP over other regions that show similar problems with ozone? (e.g., Huang et al. 2017).

- To compare with remote sensing data is a good idea; however, we decided not to perform such a comparison for three main reasons:

1) Our study focuses on surface $O_3$ and the satellite retrievals are mainly sensitive to the tropospheric column as the 0-6 km column for IASI (e.g. Safieddine et al. 2016) or the lowermost troposphere (0-3 km, see Cuesta et al., 2013).

2) To perform a proper comparison with satellites, we'd need to use their averaging kernels. To do so, we'd have to apply the averaging kernels to our $O_3$ profiles. To calculate and to store this amount of data ($O_3$ profiles) corresponding to a 10-yr period with our domain horizontal resolution would be too time and diskspace demanding.

3) Moreover, as the model levels are limited to ~20km, we'd have to complete these $O_3$ profiles by a climatology or other information for the altitudes above, in order to apply correctly the satellite AKs (e.g. see similar issues with aircraft profiles in Pommier et al., 2012). That's possible, but this climatology will have a no negligible impact on the comparison.

- Cuesta, J., Eremenko, M., Liu, X., Dufour, G., Cai, Z., Höpfner, M., von Clarmann, T., Sellitto, P., Foret, G., Gaubert, B., Beekmann, M., Orphal, J., Chance, K., Spurr, R., and Flaud, J.-M.: Satellite observation of lowermost tropospheric ozone by multispectral synergism of IASI thermal infrared and GOME-2 ultraviolet measurements over Europe, Atmos. Chem. Phys., 13, 9675-9693, https://doi.org/10.5194/acp-13-9675-2013, 2013.

- Pommier, M., Clerbaux, C., Law, K. S., Ancellet, G., Bernath, P., Coheur, P.-F., Hadji-Lazaro, J., Hurtmans, D., Nédélec, P., Paris, J.-D., Ravetta, F., Ryerson, T. B., Schlager, H., and Weinheimer, A. J.: Analysis of IASI tropospheric O3 data over the Arctic during POLARCAT campaigns in 2008, Atmos. Chem. Phys., 12, 7371-7389, https://doi.org/10.5194/acp-12-7371-2012, 2012.

- Safieddine, S., Boynard, A., Hao, N., Huang, F., Wang, L., Ji, D., Barret, B., Ghude, S. D., Coheur, P.-F., Hurtmans, D., and Clerbaux, C.: Tropospheric ozone variability during the East Asian summer monsoon as observed by satellite (IASI), aircraft (MOZAIC) and ground stations, Atmos. Chem. Phys., 16, 10489-10500, https://doi.org/10.5194/acp-16-10489-2016, 2016.

- The work done by Quennehen et al. (2016) over East Asia already gave an indication of an overestimation in $O_3$ by EMEP. However, this study and our work are difficult to compare, since the domains studied are not the same, the model version and the emissions are different and the work done by Quennehen et al. (2016) only focused on summer 2008. Moreover, no comparison with surface $O_3$ over India is presented in Quennehen et al. (2016).

- The bias in $O_3$ was already shown. We have added these sentences (in bold):

"The overestimation in $O_3$ found in this work is in agreement with previous studies (e.g. Kumar et al., 2012; Chatani et al., 2014; Sharma et al., 2016), although of course there are many differences in both emissions and models between these studies. **It has also been noted that the EMEP model slightly overestimates $O_3$, especially with the global version of the model in spring and in winter (e.g. Jonson et al., 2015b). This bias can however be impacted by the parameters used as for example the boundary conditions and the emissions. Stadtler et al. (2017) who used PANHAM anthropogenic emissions also reported an overestimation in $O_3$ over different regions such as Asia.**"

With the corresponding references:

- Jonson, J., Semeena, V., and Simpson, D., Global ozone bias Transboundary particulate matter, photo-oxidants, acidifying and eutrophying components. Status Report 1/2015, The Norwegian Meteorological Institute, Oslo, Norway, 115-128, ISSN 1504-6109, 2015b.
- Stadtler, S., Simpson, D., Schröder, S., Taraborrelli, D., Bott, A., and Schultz, M.: Ozone Impacts of Gas-Aerosol Uptake in Global Chemistry Transport Models, Atmos. Chem. Phys. Discuss., https://doi.org/10.5194/acp-2017-566, in review, 2017.

Line 255: "lack of aerosol effects"? Please clarify/expand.
The sentence has been changed (in bold):
"Some possible reasons for this might be problems with the anthropogenic and/or biogenic emissions, or over-active chemistry, e.g. over-predictions in photolysis rates for Indian conditions (**as EMEP photolysis calculations assume standard atmospheric conditions, and thus do not account for attenuation of radiation due to enhanced aerosols over polluted regions**) or problems with heterogeneous reactions."

Line 277-284: presumably there has been an increase in emission over the two different periods covered by the measurements. Could that have an impact of the comparison?
Yes, indeed. The following sentence has been added:
"It is also probable that a change in the emissions and thus in the observed $PM_{2.5}$ concentrations between the periods of both data sets has an impact on the comparison."

Line 295-300: even with more recent inventories, uncertainties in emissions persist, which could be worth noting/discussing.
We have added this sentence:
"It is also important to recall that, even with the use of recent inventories, uncertainties in emissions may persist (e.g. Saikawa et al., 2017)."

Saikawa, E., Trail, M., Zhong, M., Wu, Q., Young, C. L., Janssens-Maenhout, G., Klimont, Z., Wagner, F., Kurokawa, J., Nagpure, A. S., and Gurjar, B. R.: Uncertainties in emissions estimates of greenhouse gases and air pollutants in India and their impacts on regional air quality, Environ. Res. Lett., 12, 6, 065002, https://doi.org/10.1088/1748-9326/aa6cb4, 2017.

Line 315: as well as potential changes in vegetation in a different climate.
Yes, we have added that comment into the sentence; see our answer to your next comment.

Line 314-317: This is an important source of uncertainty. Are there any estimates in the literature of the potential magnitude of uncertainties introduced by this caveat? Is it large enough to affect the conclusion in this paper?
Yes, this is an important issue but the uncertainties cannot really be quantified or properly addressed. We have however added the following text (in bold):
"As our model does not include any $CO_2$ inhibition effect on isoprene emissions (e.g. Guenther et al., 1991; Arneth et al., 2007), **or potential changes in vegetation in a different climate**, these biogenic emissions are simply a function of temperature and increase in the FC scenarios. **The uncertainties associated with these assumptions are however difficult to quantify**. **For example, Hantson et al., (2017) found global isoprene emissions for the period 2071-2100 to be 544 TgC/yr without $CO_2$ inhibition, but only 377 TgC/yr with this effect (i.e -31%). For monoterpenes the equivalent figures were 35.7 TgC/yr and 24.8 TgC/yr (also -31%). Young et al. (2009) estimated even bigger changes for isoprene, from 764 TgC/yr to 346 TgC/yr, and showed that this uncertainty can indeed have strong effects on surface $O_3$ levels. The largest changes were found in South America and Africa, though annual**

**changes over India were only around 5-10%. Although significant, these changes are model estimates only. The experimental data behind the $CO_2$ inhibition effect are extremely limited, and as noted in Simpson et al. (2014) and reference therein, current knowledge is insufficient to make reliable predictions on this issue**."

With the corresponding references:
- Hantson, S., Knorr, W., Schurgers, G., Pugh, T. A. M., and Arneth, A.: Global isoprene and monoterpene emissions under changing climate, vegetation, $CO_2$ and land use, Atmos. Env., 155, 35-45, https://doi.org/10.1016/j.atmosenv.2017.02.010, 2017.
- Young, P. J., Arneth, A., Schurgers, G., Zeng, G., and Pyle, J. A.: The $CO_2$ inhibition of terrestrial isoprene emission significantly affects future ozone projections, Atmos. Chem. Phys., 9, 2793-2803, https://doi.org/10.5194/acp-9-2793-2009, 2009.

Line 323: could the authors include some information about the temperature change and its statistical significance in 2030 and 2050?
In the following figure, we have plotted the distribution of the temperature at 2 meter and the relative difference for both FC scenarios:

[Figure]

Note that the grey points, on the distribution of the relative difference, show the grids that do not satisfy the 95% level of significance.

We have added this information (highlighted in bold) in the text:
"This shows that for both FC scenarios, **even though the change in temperature is statistically significant (not shown),** other processes are occurring which impact on the thermal influence on the photochemical production of $O_3$."

Line 353: the three regions highlighted in the 2030 results are not the same as in 2050, but are there similar explanations? Please elaborate. Could it be that the changes seen in these quite small regions are more random, than caused by the development in climate and emissions?
We have added this sentence at the end of Section 4.2:
"The change in location of the three regions between the 2030s and the 2050s shows that the local meteorology has an impact on the change in the chemistry, such as the surface temperature. Indeed, the changes in temperature are not homogeneous over the domain and vary with the seasons."

The seasonal changes (relative difference in %) in surface temperature for the 2030s and the 2050s are plotted below:

2030s:

[Figure]

2050s:

[Figure]

Line 353-354: there is something strange with this sentence. Is "expected" the right word here, or should it be "except"?
Thank you for noticing this error. The correct word is "except". It has been corrected.

Section 4: I would like to see some discussion about the uncertainties in model representation (meteorological data) of monsoon and projected future changes, and how this could affect the results.
We have added these sentences at the end of Section "2.1. Downscaled meteorological data":
"For the future scenarios, NorESM1-M predicts an increase in temperature close to the mean of the CORDEX South Asia ensemble. For many areas there is no consensus concerning the sign of the precipitation change, except during the monsoon and the post-monsoon (October-November) in the 2050s where most of the models, including NorESM1-M, predict an increase

in precipitation over the major part of India, in comparison with the 2006-2015 period. During the pre-monsoon (April-June) in the 2050s, half of the models, including NorESM1-M, show a decrease in precipitation which is larger over the Indo-Gangetic Plains. NorESM1-M also presents this decrease in the 2030s. In winter (December-March), the western coast is characterized by an increase in precipitations, even if this change is lower in NorESM1-M than in the other models (not shown)."

These seasonal changes in precipitation can be seen in these figures:

Changes for the 2030s

[Figure]

Changes for the 2050s

[Figure]

While this following figure shows the agreement between all 8 models (including NorESM1-M) in the sign of changes in precipitation. Blue means an agreement in more precipitation,

while brown shows an agreement of less precipitation. Zero (white color) means that there are 4 models predicting an increase and 4 predicting a decrease.

[Figure]

We have also written in section 4:
"It is important to recall that uncertainties in the representation of meteorological conditions can impact our chemical results even if consistencies in the projections were simulated, especially during the monsoon and the pre-monsoon, as explained in Section 2.1"
And in Section 4.2 (in bold):
"Indeed, region (1), representing mainly a rural area, is subject to a large decrease in PM$_{2.5}$ by 8% during the monsoon. This is mainly due to the reduction in dust, representing 55% of PM$_{2.5}$, largely scavenged by the increased precipitation (+36%) **(as explained in Section 2.1).**"

Line 368:  see first comment – some information about how the model treats wet removal would be useful.
See our answer to your first comment.

Line 391:  changes in precipitation and wind speed will affect the dust production as well. Have the authors looked at this?
That is correct but while the increase in precipitation by 36% influences the decrease in dust, we have noted a slight increase in wind speed (by 1.6%) during the monsoon over region (1).

Line 401: should this be "change" instead of "variation"?
"variation" has been substituted by "change".

Section 5:  since the motivation for this paper is partly the detrimental effects of air pollution on human health, it could be interesting to also quantify changes in terms of variables such as daily maximum 8-hour concentration if high temporal resolution model data is available and to discuss PM2.5 concentrations in terms of current air quality standards. This would make a nice, policy relevant addition. Right now, the paper focuses more on the details surrounding the smaller impact of climate change, making it somewhat unbalanced.
Within UN-ECE and for integrated assessment modelling in Europe, the recommended ozone metric for health effects is SOMO35.

The SOMO35 is the indicator for health impact assessment recommended by WHO and is defined as the yearly sum of the daily maximum of 8-hour running average over 35 ppb. For each day the maximum of the running 8-hours average for $O_3$ is selected and the values over 35 ppb are summed over the whole year.

$A_8^d$ denotes the maximum 8-hourly average ozone on day d, during a year with Ny days, and SOMO35 is defined as:

SOMO35 (in ppb.days) = $\sum_{d=1}^{d=Ny} \max(A_8^d - 35\ ppb, 0\ )$

where the max function evaluates max(A−B, 0) to A−B for A > B, or zero if A ≤ B, ensuring that only $A_8^d$ values exceeding 35 ppb are included.

We have added these sentences in Section 5.1:

"This substantial increase in $O_3$ leads to a large increase in the ozone health indicator, SOMO35. The SOMO35 metric is defined as the annual sum of daily maximum running 8h average $O_3$ concentrations over 35 ppb. The SOMO35 levels for the reference scenario are already higher (Fig. S13) than over Europe (e.g. van Loon et al., 2007; EMEP Status Report 1/2017) probably related to the warmer climate and the large emissions of $O_3$ precursors over India, and the overestimation in $O_3$ from the model as shown in Section 3.1. SOMO35 is predicted to significantly increase for both FCE scenarios (Fig. S13)."

With the corresponding references:
EMEP Status Report 1/2017: "Transboundary particulate matter, photo-oxidants, acidifying and eutrophying components", Joint MSC-W & CCC & CEIP Report, 15-36, ISSN 1504-6109, 2017.
And:
van Loon, M., Vautard, R., Schaap, M., Bergstrom, R., Bessagnet, B., Brandt, J., Builtjes, P. H. J., Christensen, J. H., Cuvelier, C., Graff, A., Jonson, J. E., Krol, M., Langner, J., Roberts, P., Rouil, L., Stern, R., Tarrason, L., Thunis, P., Vignati, E., White, L., Wind, P. : Evaluation of long-term ozone simulations from seven regional air quality models, their ensemble, Atmos. Env., 41 (10), 2083-2097, doi:10.1016/j.atmosenv.2006.10.073, 2007.

And figure:

[Figure]

**Figure S13. Distribution of SOMO35 levels for the reference scenario (a), and of the relative difference in SOMO35 between the reference scenario and the FCE2030 scenario (b) and the FCE2050 scenario (c).**

Regarding $PM_{2.5}$: this is the reason there was this sentence in Section 5.2 (lines. 437-438):
"These increments alone are comparable to, or double, the annual threshold that WHO recommends not to exceed, i.e. 10 µg/m$^3$."
To clarify this point, it has been modified:
"These increments alone are comparable to the annual threshold that WHO recommends not to exceed, i.e. 10 µg/m$^3$, for the FCE2030 scenario, and the double for the FCE2050 scenario."

Line 446: please add boxes indicating regions in Fig. 16.
It has been done for Figs 13 and 16.

Line 454: Presumably, the overall PM2.5 change results in an increase in the absolute amount of EC as well, so I'm not sure about the phrasing here, i.e., "amount of EC remain low". Suggest rephrasing. Given the importance of EC/BC from a climate perspective this is an important distinction. How are PM2.5 emissions split between EC and OM in the model? (see also first comment)
Yes, the EC increases but it still represents a limited amount of $PM_{2.5}$ (3%). The sentence was confusing. Now it reads:
"It is also worth noting that even though the PPM are high for the three scenarios (close to 20% of $PM_{2.5}$), the amount of EC within these PPM remains low, around 15%."

About the split between EC and OM, please see our answer to your first comment.

Lines 449 – 454: it is interesting to note that even under increasing anthropogenic emissions, a significant fraction of PM2.5 comes from sources (dust and SOA) that are challenging, if not impossible, to control by changing policy.
This is actually a complex point. We have added these sentences at the end of Section 5.2:
"It is interesting to note that even under increasing anthropogenic emissions a significant fraction of $PM_{2.5}$ comes from sources (dust and some fraction of SOA) that are challenging to control through policy measures. Still, even biogenic, SOA is partly the product of anthropogenic emissions (and certainly land-use policy, e.g. Tsigaridis and Kanakidou, 2007, Ashworth et al., 2012), and dust is also partly a function of land-use and climate change, but such interactions are beyond the scope of our study."

And the corresponding references:
- Ashworth, K., Folberth, G., Hewitt, C. N., and Wild, O.: Impacts of near-future cultivation of biofuel feedstocks on atmospheric composition and local air quality, Atmos. Chem. Phys., 12, 919-939, https://doi.org/10.5194/acp-12-919-2012, 2012.
- Tsigaridis, K., and Kanakidou, M.: Secondary organic aerosol importance in the future atmosphere, Atmos. Environ., 41, 4682–4692, doi:10.1016/j.atmosenv.2007.03.045, 2007.

References: Quennehen, B., Raut, J.-C., Law, K. S., Daskalakis, N., Ancellet, G., Clerbaux, C., Kim, S.-W., Lund, M. T., Myhre, G., Olivié, D. J. L., Safieddine, S., Skeie, R. B., Thomas, J. L., Tsyro, S., Bazureau, A., Bellouin, N., Hu, M., Kanakidou, M., Klimont, Z., Kupiainen, K., Myriokefalitakis, S., Quaas, J., Rumbold, S. T., Schulz, M., Cherian, R., Shimizu, A., Wang, J., Yoon, S.-C., and Zhu, T.: Multi-model evaluation of short-lived pollutant distributions over east Asia during summer 2008, Atmos. Chem. Phys., 16, 10765-10792, https://doi.org/10.5194/acp-16-10765-2016, 2016.

Huang, M., Carmichael, G. R., Pierce, R. B., Jo, D. S., Park, R. J., Flemming, J., Emmons, L. K., Bowman, K. W., Henze, D. K., Davila, Y., Sudo, K., Jonson, J. E., Tronstad Lund, M., Janssens-Maenhout, G., Dentener, F. J., Keating, T. J., Oetjen, H., and Payne, V. H.: Impact of intercontinental pollution transport on North American ozone air pollution: an HTAP phase 2 multi-model study, Atmos. Chem. Phys., 17, 5721-5750, https://doi.org/10.5194/acp-17-5721-2017, 2017.